# Improved understanding of regional groundwater drought development through time series modelling: the 2018-2019 drought in the Netherlands

Esther Brakkee[1], Marjolein H.J. van Huijgevoort[1], Ruud P. Bartholomeus[1, 2]

[1]KWR Water Research Institute, Nieuwegein, 3433 PE, the Netherlands
[2]Wageningen University, Soil Physics and Land Management Group, Wageningen, the Netherlands

*Correspondence to*: Esther Brakkee (esther.brakkee@kwrwater.nl)

**Abstract.** The 2018-2019 drought in northwestern and central Europe caused severe damage to a wide range of sectors, and has made clear that even in temperate-climate countries adaptations are needed to cope with increasing future drought frequencies. A crucial component of drought management strategies is to monitor the status of groundwater resources. However, providing up-to-date assessments of regional groundwater drought development remains challenging due to the limited availability of high-quality data. This limits many studies to small selections of groundwater monitoring sites, giving an incomplete image of drought dynamics. In this study, a time series modelling-based method for data preparation was developed and applied to map the spatiotemporal development of the 2018-2019 groundwater drought in the southeastern Netherlands, based on a large set of monitoring data. The data preparation method was evaluated for its usefulness and reliability for data validation, simulation and regional groundwater drought assessment. The analysis showed that the 2018-2019 meteorological drought caused extreme groundwater drought throughout the southeastern Netherlands, breaking 30-year records almost everywhere. Drought onset and duration were strongly variable in space, with especially higher elevated areas remaining in severe drought well into 2020. Groundwater drought development appeared to be governed dominantly by the spatial distribution of rainfall and the geological-topographic setting. The time series modelling-based data preparation method was found a useful tool to enable a spatially detailed record of regional groundwater drought development. The automated TSM-based data validation improved the quality and quantity of useable data, although optimal validation parameters are probably context-dependent. The time series simulations were generally found to be reliable; however, the use of time series simulations rather than direct measurement series can bias drought estimations especially at a local scale, and underestimate spatial variability. Further development of time-series based validation and simulation methods, combined with accessible and consistent monitoring data, will be valuable to enable better groundwater drought monitoring in the future.

## 1 Introduction

In the summer of 2018, a severe drought hit large parts of northwestern and central Europe. Extremely low precipitation coincided with high temperatures, both breaking multiple-decade records in many places (see Bakke et al., 2020; Philip et al., 2020; Toreti et al., 2019). Recurring drought in summer 2019 and early 2020 worsened the situation in large parts of the area. Drying soils and declining water reserves caused damage to agricultural production and natural ecosystems, problems with

drinking water and energy production, and widespread forest fires, among other impacts (Bakke et al., 2020; Bastos et al., 2020; Buras et al., 2020; Philip et al., 2020). The kind of 'hot drought' that occurred in 2018-2019 is expected to become more frequent in the future in central and northern Europe (Philip et al., 2020; Toreti et al., 2019).

The Netherlands was one of the countries most hit by these weather extremes (Bakke et al., 2020). The damage was felt mainly in the southern and eastern parts of the country (Van de Velde et al., 2019; Van den Eertwegh et al., 2019; Witte et al., 2020b). In a country traditionally more focused on discharging water surpluses, the drought of 2018-2019 was felt by many water managers as a wake-up call, sparking a widespread search for solutions to prepare water systems for increasingly frequent drought extremes (De Lenne and Worm, 2020; IenW, 2019; Witte et al., 2020a; Van de Velde et al., 2019). In the southeastern

Netherlands, as in many other parts of the world, groundwater is a crucial water source. Accordingly, much of the damage in 2018 was directly related to deep declines in groundwater levels (LCW, 2020). Among other effects, the groundwater shortages caused severe damage in peatland and brook ecosystems (Witte et al., 2020b) and concerns over the sustainability of increased irrigation and drinking water abstractions (Van de Velde et al., 2019; Van den Eertwegh et al., 2019). Groundwater is often the most persistent water store in the landscape, reacting latest as a meteorological drought propagates into the hydrological

system (Van Loon, 2015). This makes proper management of the groundwater a crucial component of drought management strategies.

Previous studies have shown that the response of groundwater to meteorological drought can vary strongly in space. Variations in groundwater response are caused by differences in geology, water management and other catchment characteristics (Bloomfield et al., 2015; Hellwig et al., 2020; Peters et al., 2006; Van Loon and Laaha, 2015). To be able to mitigate and

prevent drought damage, it is therefore essential to understand how groundwater drought develops in both time and space. In recent years, water managers in the Netherlands have indeed expressed a need for more up-to-date, locally-specific drought information and predictions to be able to take appropriate measures (IenW, 2019; Pezij et al., 2019; Witte et al., 2020a).

Multiple recent research efforts have aimed at better understanding the variations in groundwater drought and its impacts at national and European scales (Bakke et al., 2020; Hellwig et al., 2020; Margariti et al., 2019; Van Loon et al., 2017; Brauns et

al., 2020). The 2018(-2019) drought in Europe at larger scales has so far been studied from a meteorological perspective (Bakke et al., 2020; Philip et al., 2020; Toreti et al., 2019) as well as from a hydrological perspective in, among others, Scandinavia and Switzerland (Bakke et al., 2020; Brunner et al., 2019). For the Netherlands, some assessments of the drought in the groundwater have been made based on small numbers of measurement sites and physically-based modelling studies (Van den Eertwegh et al., 2019). What is still lacking, is a more detailed image of how the 2018-2019 drought manifested

itself in the groundwater and how this varied in space, based on measurement data. This could provide valuable insights into groundwater drought dynamics and mitigation options in the Netherlands and similar groundwater-dominated lowland regions.

Groundwater heads are widely monitored in observation wells. However, analysis of groundwater drought from these data over large areas is often challenged by data quantity or quality (Kumar et al., 2016). Firstly, data usually have to be obtained from multiple organisations and contain errors and other perturbations. Secondly, the length of measured time series is often

not sufficient for drought analysis, for which at least 30-year series are recommended (Link et al., 2020). As a result, many

groundwater drought studies have focused on relatively few measurement wells with near-natural, long series or on simplified proxies (Bakke et al., 2020; Van Loon et al., 2017; Van den Eertwegh et al., 2019; Kumar et al., 2016). This may give an incomplete image of the true variability in drought dynamics. In addition, available data usually lags behind the present, hindering the up-to-date drought assessments that water managers need.

To deal with these challenges, several studies have developed methods for automated validation and lengthening of groundwater head time series (Marchant and Bloomfield, 2018; Peterson et al., 2018; von Asmuth et al., 2012). This is usually done with various types of statistical models (Peterson et al., 2018; Van Loon et al., 2017). One type of statistical modelling that has proven very useful for groundwater data is time series modelling with impulse-response functions (Bakker and Schaars, 2019; von Asmuth et al., 2002). These models describe groundwater head variations at a specific location as a function

of driving variables, usually weather data, and a fitted impulse-response function. This type of impulse-response time series modelling (TSM) allows accurate simulations to be made without a need for information on site characteristics. The simulations can be used to identify errors and other atypical behaviour in the data, and to lengthen and harmonise time series, as shown by e.g. Zaadnoordijk et al. (2019), Bartholomeus et al. (2008) and Marchant and Bloomfield (2018). As such, TSM can enable drought studies to use more observation points and to perform real-time monitoring, without the need for a complex

physically-based model. Marchant and Bloomfield (2018) were the first to develop a full time series model-based method to study groundwater drought over a large region in the UK. Although TSM-based analyses appear a valuable tool for groundwater drought assessment, their wider applicability for various cases has not yet been well explored. To be able to widely use TSM data preparation for drought studies, several questions need to be answered.

Firstly, it is not yet clear what methods are optimal for groundwater data validation. Raw groundwater data sets are usually

strongly influenced by errors and disturbances, which can hamper the reliability of analyses such as model calibration and calculation of groundwater characteristics (Post and von Asmuth, 2013; Peterson et al., 2018; Ritzema et al., 2018). Time series modelling can be used to identify time series influenced by disturbances after analysis (e.g. Marchant & Bloomfield, 2018), but also to remove irregularities from the data beforehand. In addition, TSM-based data cleaning can be combined with other more basic consistency checks, improving its effectiveness (Peterson et al., 2018). This validation method has not yet

been evaluated for the case of groundwater drought analysis.

Secondly, the reliability of time series simulations for groundwater drought analysis has not been properly tested. To understand the added value of using TSM data preparation, the gain in spatial and temporal cover needs to be balanced with a potential loss of information by cleaning and simulation. Researchers have often used TSM simulations directly as replacement of the data. This may be justified as these simulations often have a very good fit to observations (Bakker and Schaars, 2019;

Zaadnoordijk et al., 2019); however, this approach inevitably also strips out part of the external influences that are not explicitly included in the model drivers (Peterson et al., 2018; Zaadnoordijk et al., 2019). Drought occurrence and development can be strongly affected by human impacts, as well as by local-scale natural influences such as surface water influence (Margariti et al., 2019; Van den Eertwegh et al., 2019). As such, excluding such external drivers of groundwater levels may provide an incomplete image of drought dynamics (Van Loon et al., 2016). In addition, models may have intrinsic difficulties to correctly

represent groundwater behaviour during extreme drought conditions, which may cause deviating soil and groundwater flow processes (Hellwig et al., 2020; Avanzi et al., 2020). This is especially important when time series models are used for 'nowcasting' groundwater observation series. Under extreme drought conditions, this by definition involves modelling system conditions not present in the calibration period and may give incorrect results. It is therefore important to understand how the use of TSM simulations rather than measurement series affects the assessment of drought behaviour.

Given these knowledge gaps, the current study aims to evaluate the usefulness of a time series modelling-based data preparation method for regional analysis of groundwater drought. A method is developed consisting of data validation, simulation and drought assessment (Sect. 3) and applied to the 2018-2019 groundwater drought in the southeastern Netherlands, to characterise its development and recovery in time and space (Sect. 4). The usefulness of the method is evaluated by its performance and reliability with regard to groundwater data validation and simulation; and its added value for the resulting regional drought assessment (Sect. 5).

## 2 Study area and data

The study area covers roughly the southeastern half of the Netherlands (Fig. 1). This is a low-topography area above sea level, dominated by Pleistocene deposits. The study area has mainly sandy sediments at the surface, but is also partially covered by river clays and loess deposits (Fig. 1b). Elevation is mostly between 0 and 30 m AMSL, with locally higher areas (Fig. 1a). Higher elevations occur in the limestone-loess hill landscape of southern Limburg and on glacier-pushed ridges in Utrecht and Gelderland (areas indicated in Fig. 1a). Land use is dominated by agriculture, while the glacial ridges are covered mainly with forest. The area has a temperate climate with a yearly precipitation surplus ($P$ 700-950 mm j$^{-1}$, $ET_{ref}$ around 600 mm j$^{-1}$). In addition, the groundwater system is affected by abstractions for drinking water and irrigation, as well as by drainage systems in the lowest-lying parts of the study area.

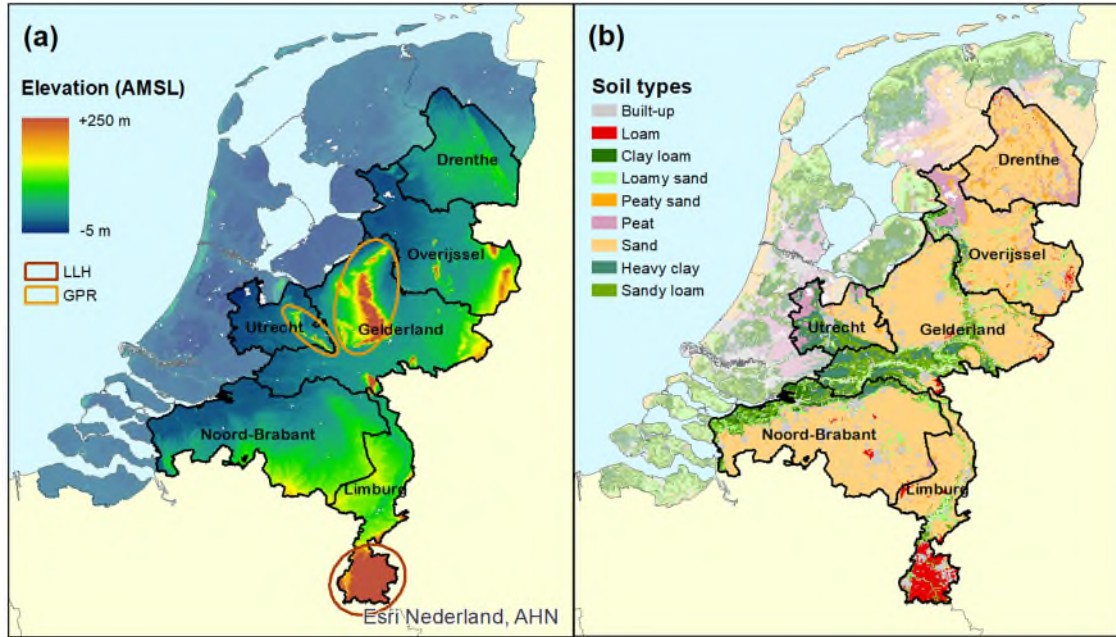

**Figure 1: Study area. a: Elevation (AHN, 2019); b: Soil types (WUR, 2006). The higher elevated limestone-loess hill landscape in southern Limburg (LLH) and glacier-pushed sand ridges (GPR) are indicated.**

Groundwater head data were supplied by several regional water managing bodies. For some areas, additional series were obtained from the Dutch national groundwater database DINO (TNO). The data consist of groundwater head time series with a twice-monthly to sub-daily frequency, mostly running until spring 2019. In addition, metadata of the monitoring wells were available, including location, filter depth and surface level. Only data from the first filter of boreholes was used, generally representing the phreatic level. Those series were selected that contained > 10 years of consecutive data to ensure sufficient data for time series model fitting (see Zaadnoordijk et al. (2018)); and ended after 2018-08-31. The 2018-2019 meteorological drought peaked in summer 2018 (Fig. 3); therefore summer 2018 was chosen as the focus period for model evaluation. This resulted in 2722 series for further analysis.

Daily precipitation ($P$) and reference evapotranspiration ($ET_{ref}$) were obtained from the Royal Netherlands Meteorological Institute (KNMI) for January 1990 to May 2020 (KNMI, 2020). Data were used from 15 general weather stations (for $ET_{ref}$) and 114 precipitation stations distributed homogeneously over the study area. Reference evapotranspiration is determined by KNMI following Makkink (1957). The $ET_{ref}$ series did not always cover the full period of interest; gaps were filled with the nearest station that did have full data (maximum distance around 50 km).

## 3 Methods

The method for data preparation and drought analysis consists of three components (Fig. 2): 1) validation of the observed groundwater heads, 2) simulation of groundwater heads, and 3) conversion to a standardised groundwater index (SGI). Each

step is evaluated by one or more tests. In addition, the resulting drought assessment for the case study region is explored as an example of a regional-scale application.

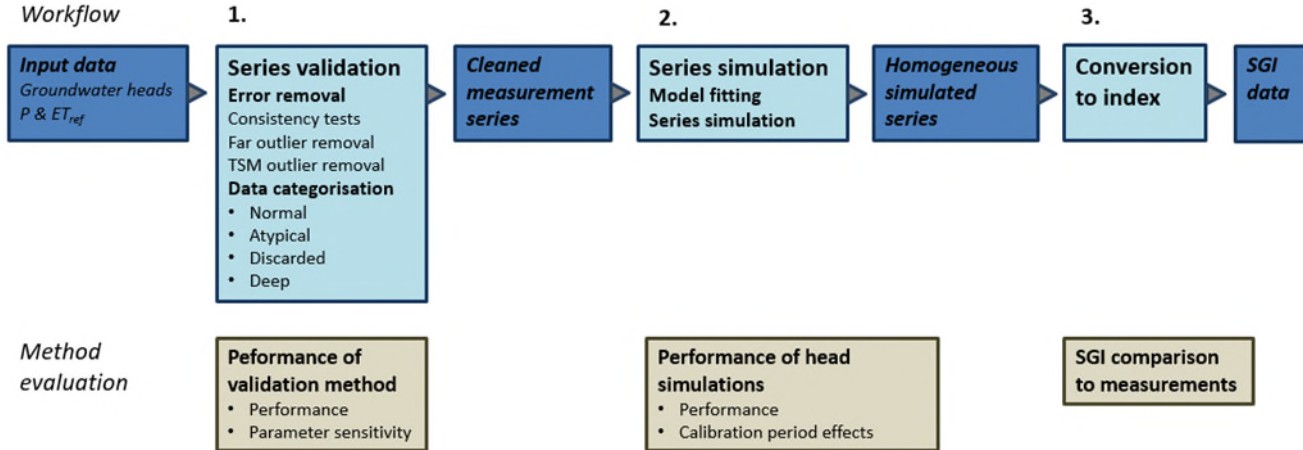

Figure 2: Method workflow.

### 3.1 Time series modelling method

This study has made use of time series modelling with predefined impulse-response functions, as developed by von Asmuth et al. (2002). Time series modelling was done with the Pastas package for Python developed by Collenteur et al. (2019). The used model setup largely follows Collenteur et al. (2019) and is described in more detail in Appendix A. In short, groundwater levels were simulated as a base level $d$, overlain by a temporal fluctuation in response to external stresses – in this case only recharge. Here, recharge is estimated as:

$$R(t) = P(t) - f \cdot ET_{\mathrm{ref}} \tag{1}$$

With $f$ a calibration parameter. The use of the linear recharge model of Eq. 1 is a simplification that may not be optimal for all locations in this study (Collenteur et al., 2021; Bakker and Schaars, 2019). However, as linear recharge models have been successfully applied in many cases in the Netherlands (e.g. Zaadnoordijk et al., 2019; von Asmuth et al., 2012) and as we aimed to explore the potential of impulse-response time series modelling for drought studies rather than comparing different model setups, we chose to use the simplest model setup possible; see section 5.1.2 for further comments.

The response of groundwater to a recharge impulse is modelled by a scaled gamma function as in von Asmuth et al. (2002) and Collenteur et al. (2019) (see Appendix A). The variation in groundwater heads over time is calculated by convolution of this impulse-response function with the recharge time series.

This gives five parameters to be calibrated for each individual location. Parameter $A$ represents the long-term response of the groundwater level to a constant recharge input of one unit, in this case 1 mm; $a$ and $n$ determine the shape of the recharge response function; $f$ is the influence of $ET_{\mathrm{ref}}$ relative to precipitation; and $d$ is the groundwater base level. For purposes of parameter calibration, also an exponential noise model is fit to the residuals with the additional noise decay parameter $\alpha$.

Table A-1 gives the calibration settings used for each parameter. The default method for parameter optimisation was used, minimising the weighted squared noise using a least squared method (Collenteur et al., 2019).

## 3.2 Series validation

Raw groundwater data sets are usually strongly influenced by errors and disturbances. Often no information is available on potential sources of deviations, so that these have to be identified from the groundwater data itself (Post and von Asmuth, 2013; Peterson et al., 2018; Ritzema et al., 2018). Phreatic groundwater levels typically follow an annual cycle, overlain by faster fluctuations in response to rainfall and evapotranspiration. An actual series of measured groundwater heads will often show deviations from this expected pattern. Deviations may be of short duration, such as caused by a typing error, temporary

instrument failure or short-term groundwater abstraction. These can be denoted outliers: a small number of measurements far from the expected level, occurring over a short period (days or weeks) relative to the general (seasonal) fluctuations in most groundwater series (e.g. Peterson et al., 2018). Deviations may also be structural, affecting the series behaviour over months or longer. These are visible as level shifts, trends and other abnormal patterns in the data series (see Appendix A3). Such long-term deviations can be caused by errors, such as instrument drift; however, they can also reflect real groundwater behaviour

caused by local natural or human influences, such as surface water influence or abstraction (Post and von Asmuth, 2013; Zaadnoordijk et al., 2019; Margariti et al., 2019). Log book notes from data collectors available for a small subset of our dataset indeed showed frequent disturbances such as short-term abstractions, changes in water management, sensor problems, relocation of wells, well maintenance and other issues.

To prepare the data for the drought analysis, we used a validation setup that treats short-term (outliers) and long-term deviations

separately. The validation method aims to remove all important outliers: erroneous outliers will lead to incorrect conclusions on the occurrence of extremes, while real short-term disturbances in the groundwater heads are also less relevant for understanding the slow-developing impacts of drought, which is generally considered to occur on timescales of months to years (van Loon, 2016). Erroneous *long-term* deviations are also undesirable for drought analysis, as these disturb the groundwater level distribution on which drought thresholds are based (see Sect. 3.3). However, *real long-term* deviations in

the groundwater level, such as caused by long-term abstraction and land use effects, should ideally be retained to capture the real variability in drought behaviour (Van Loon et al., 2016). Whether atypical behaviour in a series is caused by errors or by real external influences is very difficult to distinguish by automated methods. Our approach is therefore to classify the series according to their long-term behaviour; this allows for retaining some of the potentially influenced series in the analysis, while acknowledging their lower reliability.

The outlier cleaning consisted of the following steps (see Table 1 for parameters):
1. Basic metadata consistency check. Measurements below the well filter or $> TH_{inund}$ above the surface level were removed, as these are likely to point to erroneous measurements or metadata.

2. Removing far outliers by range. As a fast first cleaning step, far outliers were identified by isolating the top and bottom fraction of the measurement range ($F_{range}$) and identifying them as outliers if their removal caused a reduction in range of > $TH_{red}$.

3. Outlier removal by time series modelling. A model was fit for each series using precipitation and $ET_{ref}$ from the nearest weather stations. All measurements outside a range of $n_{sd}$ times the standard deviation of the residuals around the simulation were removed. This step was repeated $n_{iter}$ times to deal with outliers disturbing the model fitting (Peterson et al., 2018; Leunk, 2014).

For the long-term behaviour classification, a new time series model was fit on the resulting cleaned series. The explained variance percentage (EVP, equal to $r^2 \cdot 100$) and model parameters $A$, $f$ and $d$ were saved. In addition, a linear trend was fitted through the residual series and the $p$-value and $r^2$ of the trend were saved. Finally, the series were checked for consecutive periods with missing data of > 4 years which would hamper the required 10-year data period (Sect. 2) and, if these were present, only the time period after the last data gap was used. Based on these indicators the series were ordered into four categories of long-term behaviour (see Appendix A3 for examples):

1. Discarded series: series with very strong deviations from the expected behaviour, as indicated by EVP < $TH_{EVP}$; or insufficient data for analysis (< 10 years, data gaps > 4 years, or no data over June-August 2018).

2. Deep-groundwater series: mean water table depth (WTD) > 5 m. These series typically showed a very slow, smoothed behaviour and often poor model fit; the used validation method is probably less suitable for these series.

3. Atypical series: series with mild deviations: EVP >= $TH_{EVP}$, but containing a trend or atypical parameters. This points to potential errors, external (human) influence or groundwater processes that deviate from the TSM assumptions used in this study. Locations were marked as atypical if they had a trend in the residuals with $p < 0.05$ and $r^2 > TH_{r2}$; or unusual values for the $f$ and $A$ parameters ($TH_f$ and $TH_A$).

4. Normal series: EVP >= $TH_{EVP}$, no other issues.

The cleaned measurement series of the normal, atypical and deep categories were aggregated to daily means and saved for further use. Table 1 shows the validation parameters used for this study. Several parameters were chosen by initial trial-and-error testing; the sensitivity to these parameters was tested as explained below.

**Table 1: Used parameters for the series validation.**

| | Parameter | | Used value | Justification |
|---|---|---|---|---|
| Outliers | MetaCheck | Metadata check performed | yes | |
| | $TH_{inund}$ | Maximum allowed inundation | 0.2 m | Shallow inundation possible in study area, deep inundation unlikely (Leunk, 2014) |
| | $F_{range}$ | Fraction of range identified as potential outlier in far outlier cleaning | 0.2 | Tested |
| | $TH_{red}$ | Minimum range reduction (fraction) to remove outliers in far outlier cleaning | 0.5 | Tested |

| | | | | |
|---|---|---|---|---|
| | $n_{sd}$ | Threshold number of standard deviations to remove outliers | 4 | Tested |
| | $n_{iter}$ | Number of iterations in TSM outlier cleaning | 2 | Tested |
| Long-term classification | $TH_{EVP}$ | EVP threshold to discard series | 60 % | Visually estimated as suitable; see Appendix A2 |
| | $TH_{r2}$ | $r^2$ threshold of trend in residuals to mark series as atypical | 0.15 | Tested |
| | $TH_f$ | Threshold in $f$ value to mark series as atypical | > -0.05 or < -1.95 | Close to parameter bounds; see Table A1 |
| | $TH_A$ | Threshold in $A$ value to mark series as atypical | >1.5 | Far from normal range of values for the given dataset; see Table A1 |

The performance of the validation method was evaluated on a test set of 180 randomly selected series (30 from each province). These series were visually checked for the occurrence of 1) outliers (series to clean); 2) serious long-term deviations such as level shifts or strong trends (series to discard); and 3) milder long term-deviations such as lighter trends (series to mark as atypical); see figures A1-4 for examples.

The validation routine was applied to the test set with the standard parameters of Table 1; and with 20 alternative parameter
sets (Table A2). In set 2-11, the parameters were varied individually to more conservative (less cleaning and discarding) and more rigorous values (more cleaning and discarding). In set 12-19, combinations of conservative/rigorous outlier cleaning and long-term deviation identification parameters were tested; and in set 20 and 21 versions are tested with only TSM-based outlier cleaning (no basic cleaning step) and no outlier cleaning at all.

The validation results from all parameter sets were compared to the visual validation and scored by 1) correct identification of
outliers (cleaned if needed); 2) correct identification of serious long-term deviations (discarded if needed); and 3) correct identification of mild long-term deviations (marked as atypical if needed). Scoring was done as True or False Positive (deviations correctly recognised) and True or False Negative (absence of deviations correctly recognised) for the three categories.

### 3.3 Series simulation

The cleaned series from the validation step were used to simulate groundwater heads for the drought analysis. It was chosen to use simulated series, rather than interpolating the measurement series with TSM (Marchant and Bloomfield, 2018), to ensure regular series without sudden level shifts and prevent influence of remaining outliers. A model was fit on the cleaned measurement series (see section 3.1 and appendix A1); and for those models with an EVP > 60% (Table 1), a daily-step groundwater head series was simulated for the period of interest, in this case 1 January 1990 to 31 May 2020. As most
measurement series originally ran until spring 2019, roughly one year of 'nowcasting' was added to the data.

The performance of the simulations was assessed by the root mean squared error (RMSE) and the mean error (ME) of the simulated groundwater levels over the full length of the measurement series. ME was calculated as $ME = mean(GWL_{sim} -$

$GWL_{obs}$) to quantify bias. To assess model performance under dry conditions specifically, the measurements of each series were subdivided into 'low-head periods' (measured head < 20th percentile), 'medium-head periods' (20th to 80th percentile)

and 'high-head periods' (> 80th percentile), and the RMSE and ME were re-calculated for these periods. Finally, RMSE and ME were determined specifically over July-November 2018, the period in which the meteorological and hydrological droughts peaked. All performance measures were expressed both as the absolute value in meters, and as fraction of the mean water table depth (WTD) at the location. The latter measure may give a better image of the scale of the errors, as the impact of a small change in groundwater head on vegetation and hydrological processes is generally much larger where groundwater levels are

normally shallow (Bartholomeus et al., 2012; Witte et al., 2020b).

The sensitivity of the simulation of drought conditions to the calibration period was also tested. Models were re-calibrated on the period until the start of the 2018 growing season (1 April 2018) and the groundwater behaviour for the rest of 2018 was 'nowcasted' with the available weather data. Simulation performance was again valuated by the RMSE and ME over July-November 2018, and compared with the simulations calibrated on the full period.

The groundwater behaviour at an individual measurement location can be summarised through the recharge response time, which can be derived from the fitted time series models. The response time was here defined as the time after which 50 % of the groundwater head response to a recharge event has occurred (e.g. Zaadnoordijk et al., 2019). It was derived by intersecting the step response function obtained from Pastas (function get_step_response) with the line 0.5$A$. These response times are a characteristic of the full groundwater series, not just the drought period. Still, the response times and their spatial distribution

give a further indication of the validity of the time series models and the drivers of groundwater drought development.

### 3.3 Drought index

To identify drought periods in time series of hydrological variables, and to compare drought severity between locations, standardised drought indices are used. We quantified the development of meteorological drought over 2018-2020 by the three-month-aggregated Standardised Precipitation Evaporation Index (SPEI) (Vicente-Serrano et al., 2010). The study area was

divided into four zones (see Fig. 2); the SPEI-3 for each zone was calculated from the average precipitation of all weather stations within the zone and the distance-weighted mean $ET_{ref}$ of the three stations closest to the midpoint. This midpoint method was necessary to obtain representative values for each zone from the relatively few evapotranspiration stations present (15). SPEI was calculated by a normal distribution transformation for simplicity; the three-month accumulation time was chosen because this most clearly showed the meteorological droughts at a time scale comparable to the variations in

groundwater level.

For groundwater heads the Standardised Groundwater Index (SGI) was applied  (Bloomfield and Marchant, 2013). The SGI method consists of transforming a measured groundwater head series at a specific location to a standard normal distribution; this produces a drought index series varying roughly between -3 and 3, indicating conditions from extremely dry to extremely wet compared to the normal situation. When analysing drought indices for multiple locations, a common reference period must

be used. It is generally recommended to use a period of at least 30 years to ensure a proper estimation of the long-term "normal

situation" (McKee et al., 1993; Van Loon et al., 2016; Ritzema et al., 2018). Here, the period January 1990 - December 2019 is used throughout as the reference period.

For precipitation, the transformation step is usually done by fitting a gamma distribution function to the data (McKee et al., 1993). For groundwater heads, however, distribution shapes vary widely between locations (Bloomfield and Marchant, 2013; Dawley et al., 2019; Loáiciga, 2015). Fitting individual parametric distribution functions to each location based on the 30-year monthly series used here is likely to give unreliable results; for example, Link et al. (2020) find that in most cases more than 100 data points are needed for fitting reliable parametric distributions on hydrological series, while incorrect transformations give a high risk of biased drought index values (Svensson et al., 2017). Groundwater levels were therefore transformed using a normal scores transform (see e.g. Bloomfield and Marchant, 2013). This is a nonparametric transformation method that has the advantage of being simple and transparent and circumvents the risk of bias due to erroneous distribution fits. For each location, the simulated series was first aggregated to monthly mean levels. Transformation was then done separately for each calendar month. For each calendar month with $n$ years of data, in this case 30, cumulative probability values are taken, uniformly spaced over the interval $(1/2n)$ to $(1 - 1/2n)$; the corresponding SGI values are found by applying an inverse cumulative distribution function to these values. The resulting SGI values are assigned to the groundwater head measurements of the given calendar month by their rank from low to high. This method of calculating SGIs results in a limited number of 'discrete' SGI values that correspond directly to the rank of the groundwater level compared to the rest of the reference period. Table 2 gives the SGI values and their corresponding rank and drought severity in this study. The drought severity classes follow the classification by McKee et al. (1993) that has been frequently used in drought studies. Note that the SGI, unlike the SPI, is not aggregated over multiple months; the SGI values given here therefore represent 'SGI-1'.

**Table 2: Used categories for the standardised groundwater index, with the corresponding groundwater level rank in a 30-year record.**

| SGI | Drought category | Rank (dry → wet) |
|---|---|---|
| **> 0** | No drought | >15 *(wettest 15 years)* |
| **0** to **-1** | Mild drought | 6 – 15 |
| **-1.5** to **-1** | Moderate drought | 3 – 5 |
| **-2** to **-1.5** | Severe drought | 2 |
| **< -2** | Extreme drought | 1 *(driest year)* |

The SGI values for the months outside the reference period (January 2020 – May 2020) were estimated by linearly interpolating the groundwater head-SGI relation for the calendar month. If the heads fell outside the range reached in the reference period, they were assigned the most extreme SGI value.

To test how the use of TSM simulations rather than measurement series affects drought analysis, SGI values were also calculated directly for a selection of locations that had long measurement series. To collect enough series for comparison while preventing the influence of differing time periods, a minimum of 27 years was used. All series with at least 27 years of data

(starting before 1 January 1993) were selected from the cleaned measurement series, resulting in 531 series. The SGI values for 2018 were calculated in the same way as for the simulated series, with the SGI of a calendar month calculated only if at least 25 years of data were available. In some cases the number of data points will thus be some smaller than for the simulation-based SGI. However, an $n$ of 25 instead of 30 does not affect the classification for the lowest drought categories as shown in Table 2; also an exploratory test (not shown) indicated that the slight mismatch in period did not affect the patterns in the resulting comparison. The simulation-based and measurement-based SGI values were compared by regression (Spearman's $\rho$).

## 4 Results

### 4.1 Usefulness of TSM method for drought data validation and simulation

#### 4.1.1 Validation performance

The validation performance for the standard validation parameter set is shown in Table 3. For a large majority of test series the routine performed a correct action with regard to cleaning outliers, discarding strongly deviating series and marking atypical series. In the outlier cleaning, there is a relatively large fraction of false positives (removal of non-existing outliers). This mainly occurs for points in well-modelled series, without affecting the character of the series or the drought extremes. The false negatives (outliers not recognised) partly did concern influential outliers; another relatively frequent problem was the incomplete removal of a group of outliers (not shown in Table 3). With regard to the strong long-term deviations, there is a relatively large fraction of false positives (unduly discarded series). This is partly caused by the incomplete outlier cleaning for some of the series. The number of series marked as atypical by both the visual and automated validation is relatively small. This category is hard to identify consistently by visual inspection, which may explain the false positives and negatives.

Table 3: Validation performance for the standard parameter set (Table 1). Number of series with insufficient data is 19, so $n_{total}$=161. True Positive=identified in both manual and automatic validation; True Negative=not identified in either manual or automatic validation; False Positive=identified in automatic validation but not in manual validation; False Negative=identified in manual but not in automated validation. Excl. deep: excluding deep-GWL series. Last column: percentage of series with outliers and long-term deviations correctly or reasonably identified.

| Outliers | True Positive | True Negative | False Positive | False Negative | False Negative excl. deep | Correct clean action |
|---|---|---|---|---|---|---|
| | 71 (44%) | 56 (35%) | 26 (16%) | 8 (5%) | 6 (4%) | **79 %** |
| **Strong long-term deviations** | True Positive | True Negative | False Positive | False Negative | False Positive excl. deep | Correct discard action |
| | 32 (20%) | 103 (64%) | 26 (16%) | 0 (0%) | 22 (14%) | **84 %** |
| **Mild long-term deviations** | True Positive | True Negative | False Positive | False Negative | False Positive excl. deep | Correct mark action |
| | 7 (4%) | 131 (81%) | 11 (7%) | 12 (7%) | 7 (4%) | **86 %** |

The parameter sensitivity test (see Table 4 and Appendix A2) showed that the standard parameter set performed relatively well in comparison with other parameter sets. The sensitivity to the parameters of the far outlier cleaning ($F_{range}$, $TH_{red}$) and the

number of iterations in the TSM outlier cleaning ($n_{iter}$) was low, while the standard deviation range for the TSM cleaning ($n_{SD}$) and the thresholds for discarding and marking series ($TH_{EVP}$ and $TH_{r2}$) did have a large effect. Applying an outlier cleaning step in general increased the simulation performance (mean EVP 60 to 62 %, set 1 vs 21) and allowed more series to be retained for analysis (60 % to 64 % of series), with the TSM cleaning being responsible for most of the outlier cleaning. Taking a conservative low EVP threshold appears to give a good performance on the strong deviation identification (set 8 and 16), but the number of false negatives is high, while a strict EVP threshold of 80 % causes a majority of series to be discarded. Changing the threshold on the residual trend to mark series as atypical (set 10 and 11) caused either almost all or almost no series to be marked and did not substantially improve the performance.

**Table 4: Summary results of validation parameter sensitivity test. Nr Discard/Atypical: number of series discarded/marked as atypical. Cleaning/Discard/Marking correct %: percentage of series with correct action identified for cleaning, discarding and marking. See Table A1 for explanation of the parameter sets.**

| Set | Name | Mean EVP [%] | Nr Discard | Nr Atypical | Cleaning correct % | Discard correct % | Marking correct % |
|---|---|---|---|---|---|---|---|
| *1* | *Standard* | *61.9* | *58* | *18* | *79* | *84* | *86* |
| 2 | FarOutliersConservative | 62 | 58 | 18 | 80 | 84 | 86 |
| 3 | FarOutliersRigorous | 61.9 | 58 | 18 | 79 | 84 | 86 |
| 4 | OutliersConservative | 60.6 | 64 | 17 | 80 | 81 | 86 |
| 5 | OutliersRigorous | 68.8 | 41 | 28 | 60 | 86 | 82 |
| 6 | IterationsConservative | 61.7 | 61 | 18 | 79 | 82 | 86 |
| 7 | IterationsRigorous | 62.2 | 58 | 18 | 79 | 84 | 86 |
| 8 | EVPConservative | 61.9 | 29 | 33 | 79 | 86 | 80 |
| 9 | EVPRigorous | 61.9 | 119 | 4 | 79 | 47 | 88 |
| 10 | TrendConservative | 61.9 | 58 | 7 | 79 | 84 | 88 |
| 11 | TrendRigorous | 61.9 | 58 | 39 | 79 | 84 | 74 |
| 12 | OutliersConservative_EVPConservative | 60.5 | 29 | 30 | 80 | 86 | 82 |
| 13 | OutliersConservative_EVPRigorous | 60.5 | 122 | 4 | 80 | 45 | 88 |
| 14 | OutliersConservative_TrendConservative | 60.5 | 66 | 7 | 80 | 80 | 88 |
| 15 | OutliersConservative_TrendRigorous | 60.5 | 66 | 36 | 80 | 80 | 75 |
| 16 | OutliersRigorous_EVPConservative | 71.1 | 15 | 45 | 59 | 89 | 75 |
| 17 | OutliersRigorous_EVPRigorous | 71.1 | 82 | 8 | 59 | 68 | 87 |
| 18 | OutliersRigorous_TrendConservative | 71.1 | 36 | 13 | 59 | 87 | 85 |
| 19 | OutliersRigorous_TrendRigorous | 71.1 | 36 | 45 | 59 | 87 | 72 |
| 20 | Standard_TSMcleaningOnly | 62.1 | 57 | 19 | 79 | 84 | 86 |
| 21 | Standard_NoOutlierCleaning | 59.9 | 64 | 16 | 68 | 81 | 88 |

When applied to the full study dataset, the validation procedure discarded 31 % of the groundwater head measurement series, so that 1869 of the original 2722 series remained for analysis. A poor model fit was the most frequent cause for discarding. 10 % of the series were maintained as atypical series, while another 12 % of locations had a deep groundwater table, being less suitable for the used validation method.

### 4.1.2 Simulation performance

In the simulation step, 1632 locations were modelled with sufficient quality (EVP > 60 %). Overall, these series were simulated with an average error of 14 cm, resulting on average in a 20 % error in the groundwater table depth (Table 5). The bias is low with -1 mm. Subdivision into dry, normal and wet conditions (0-20[th], 20-80[th] and 80-100[th] percentile of groundwater levels, respectively) shows that the errors are larger for more extreme groundwater levels (dry and wet conditions). More precisely, the models tend to underestimate the extremes: there is a positive average bias in periods of low groundwater levels, and a negative bias when groundwater levels are in their high ranges. There was no clear spatial pattern in this bias. Also during the main period of groundwater drought in July-November 2018, the simulation error is above average with 18 cm, but there is only a small negative bias of 1 cm.

**Table 5: Performance of the groundwater head simulations for the full simulation period and summer 2018. RMSE=Root Mean Squared Error, ME=Mean Error of simulations versus measurements, given in meters and as fraction of the mean groundwater table depth (WTD).**

| | RMSE | | ME | |
|---|---|---|---|---|
| | Mean value *(range)* [m] | Fraction WTD [-] | Mean value *(range)* [m] | Fraction WTD [-] |
| **Full period** | 0.14 *(0.03…1.7)* | 0.20 | $-1.2 \cdot 10^{-3}$ *(-0.3…0.2)* | $8.1 \cdot 10^{-3}$ |
| **Dry** | 0.16 *(0.03…2.0)* | 0.25 | 0.076 *(-0.7…0.6)* | 0.15 |
| **Normal** | 0.12 *(0.02…1.6)* | 0.18 | $-3.3 \cdot 10^{-3}$ *(-0.3…0.4)* | 0.039 |
| **Wet** | 0.15 *(0.03…1.7)* | 0.20 | -0.070 *(-1.3…0.3)* | 0.089 |
| **Jul-Nov 2018** | 0.18 *($6.0 \cdot 10^{-3}$…3.2)* | 0.29 | -0.010 *(-2.9…1.5)* | 0.22 |
| ***Jul-Nov 2018, calibrated until 1 April 2018*** | 0.20 *($1.0 \cdot 10^{-3}$…3.7)* | 0.31 | -0.015 *(-3.5…1.2)* | 0.24 |

### 4.1.3 Calibration period sensitivity

In addition to the fully calibrated simulations, the sensitivity of the model simulations during drought to the used calibration period was tested (Table 5, last row). When the 2018 drought summer was simulated with a TSM model calibrated until spring 2018, the average error in the predicted groundwater heads was 20 cm, giving a relative error in the groundwater depth of 31 %, thus performing slightly poorer than the fully calibrated simulations (error of 20 %). Similar to the fully calibrated simulations, there is a (small) negative mean error, indicating that the declines in groundwater level over summer, and thus the severity of drought, are slightly overestimated. This means that, as expected, the simulation of groundwater levels under extreme drought conditions outside the range of conditions in the model calibration has a relatively low reliability. However, the difference in average error is only 0.03 m, indicating that the effect is relatively small compared to the error already present in the fully calibrated simulations. There are no clear spatial patterns in the RMSE of the simulations (not shown). The sensitivity to the calibration period thus appears independent of specific catchment characteristics in the study area.

**4.2 Development and recovery of the 2018-2019 groundwater drought in the Netherlands**

The groundwater drought of 2018-2019 was driven by exceptionally dry weather conditions. The meteorological drought started in spring 2018 and peaked in late summer (Fig. 3). After a relatively normal winter, summer 2019 again showed moderate to severe drought. The winter of 2019-2020 was relatively wet, but exceptionally low rainfall in spring 2020 caused a return to extreme meteorological drought conditions. The meteorological drought was not spatially uniform. Especially in spring 2018 and summer 2019, the western part of the study area experienced less dry conditions than the east (Fig. 3).

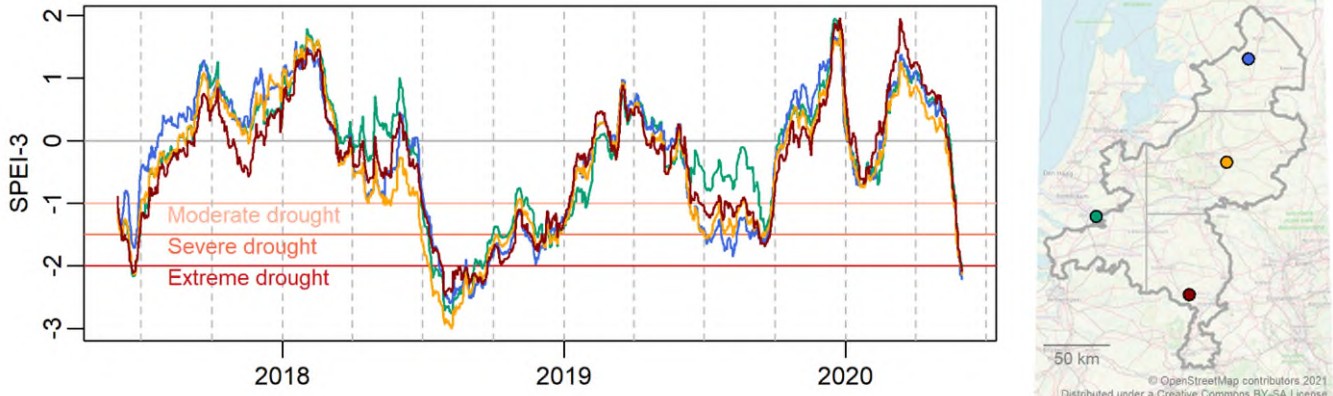


**Figure 3: Development of the meteorological drought over 2018-2020 in the west, north, mid-east and southeast sections of the study area given by the 3-month SPEI. The map shows the four sections.**

The development of the groundwater drought in the southeastern Netherlands over 2018 is visualised in Figures 4 and 5. The 2018 growing season started with uniformly normal to high groundwater levels over the study area (Fig. 4). Drought started

developing in May and June, with drought onset varying between locations. By July and August, severe to extreme groundwater drought occurred over most of the area. In September, heavy rain in the west of the study area slightly alleviated the drought conditions. However, the drought situation worsened again in autumn, reaching its height in October and November when the simulations show almost uniform extreme drought over the study area. By December, a slow recovery is visible.

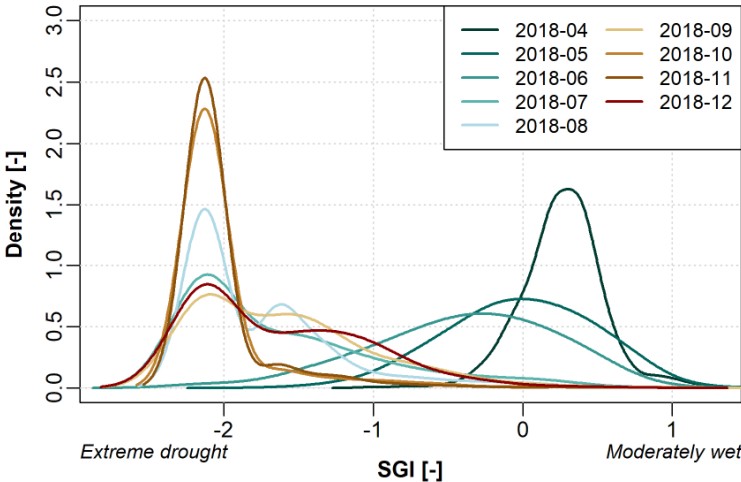


**Figure 4: Monthly distributions of the Standardised Groundwater Index over 2018 in the southeastern Netherlands.**

The simulated data show distinct spatial patterns in the development of the groundwater drought (Fig. 5). Especially southern Limburg and the ridges in Utrecht and Gelderland with their distinct geology and topography (see Fig. 1) reacted more slowly than the rest of the study area and did not experience drought conditions yet in 2018. Also the fast recovery of the low-lying

western Utrecht area in autumn stands out.

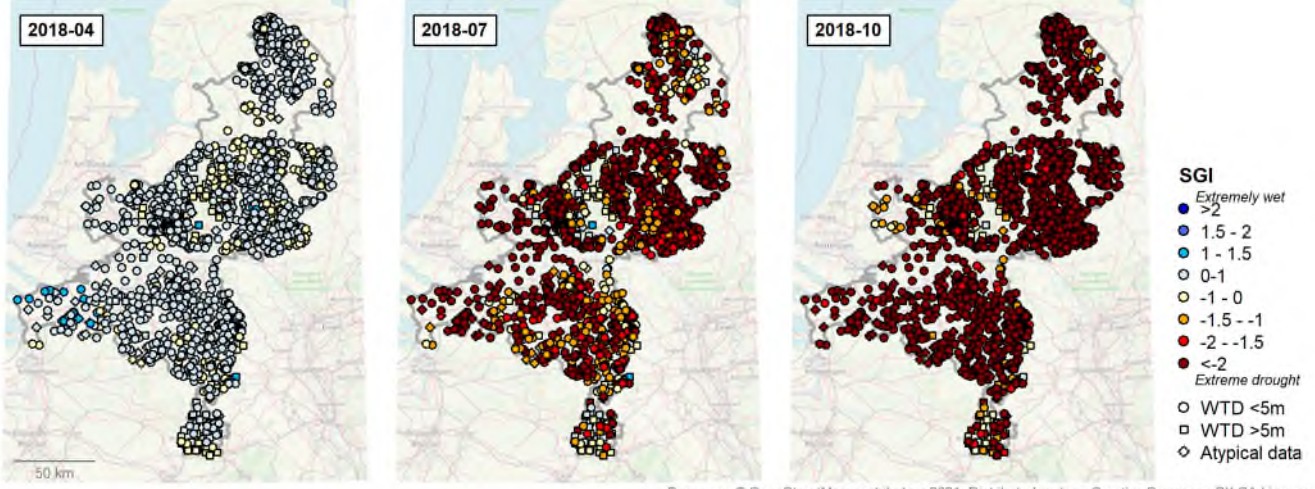

**Figure 5: Groundwater drought development in 2018: SGI of simulated series. WTD: Water table depth.**

The simulations over 2019-2020 show that also drought recovery was strongly variable in space (Fig. 6). In spring 2019, groundwater heads in the west of the study area were again approaching normal levels, but the eastern regions had recovered

poorly. By this time extreme groundwater drought had also developed on the high ridges and in southern Limburg. The summer of 2019 again brought severe to extreme groundwater drought, this time clearly concentrated in the east of the study area,

corresponding to the differences in meteorological drought. In March 2020, groundwater levels had returned to relatively high levels. However, the exceptionally dry weather in April rapidly resulted in a new severe drought situation by May.

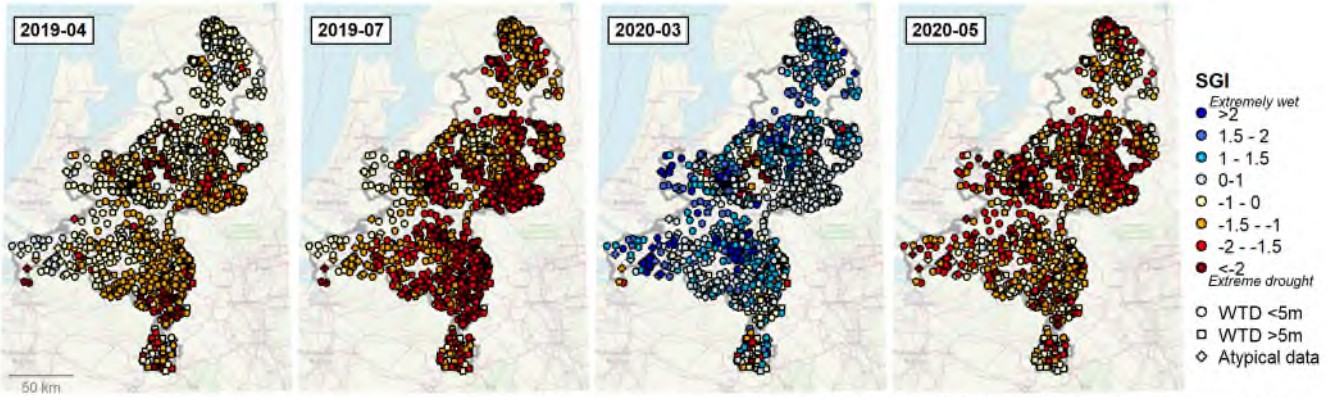

Figure 6: Groundwater drought over 2019-2020: SGI of extended groundwater series. WTD: Water table depth.

The groundwater response times derived from the time series models are shown in Fig. 7. The response times range from a few days to up to two years, but are generally relatively short: 94 % of locations has a response time of less than one year. The spatial distribution of the response time corresponds with the topography and the occurrence of glacial sand ridges in the landscape (Fig. 1) and with the propagation speed of meteorological drought to groundwater drought in 2018-2020 (Fig. 5, 6).

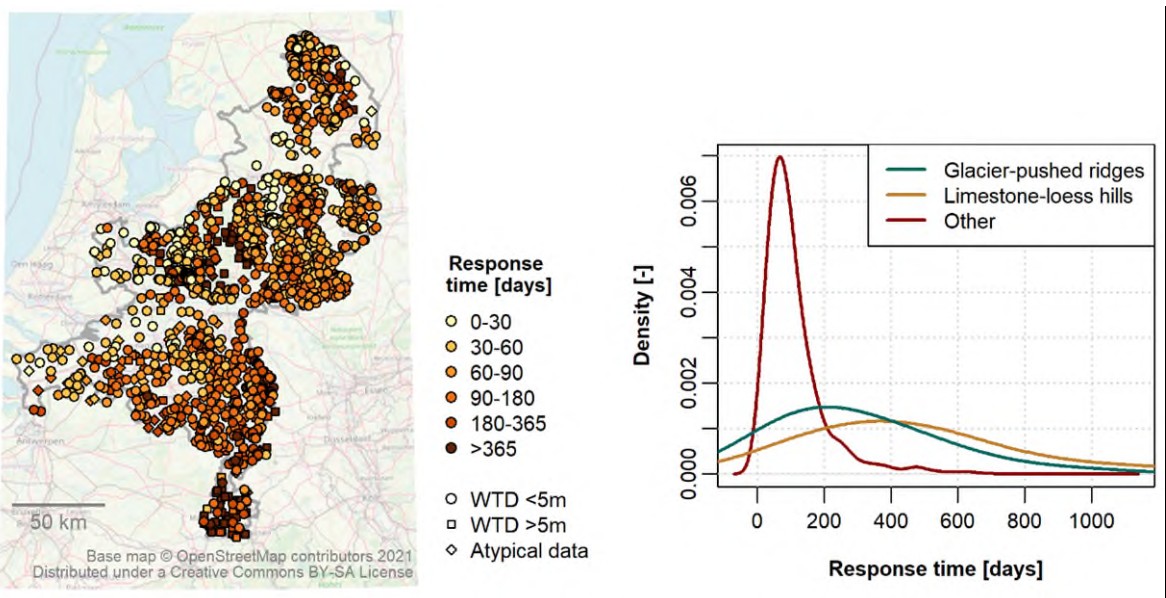

Figure 7: Response time to recharge as derived from the fitted response functions. See Fig. 1 for the location of subregions.

### 4.3 Usefulness of TSM method for regional drought assessment

For a subset of the locations ($n$ = 531), a long groundwater measurement series was available and the SGI values resulting from the simulated series could be compared to those obtained directly from the cleaned measurement series (Fig. 8). The

comparison shows that the simulations follow the same general drought behaviour as the measurement series, as the two follow a 1:1 line (Spearman's $\rho = 0.8$). However, the simulations generally show a smaller spatial variation than the measurements. This is also visible in measurement-based SGI maps (Fig. 9), which show more scatter and local extremes than the simulation-based drought maps. In addition, the simulations for 2018 tend to slightly overestimate drought severity in the low ends of the drought (lower left in Fig. 8). This is contrary to the general tendency of the head simulations towards positive bias during

drought periods, shown in Table 5.

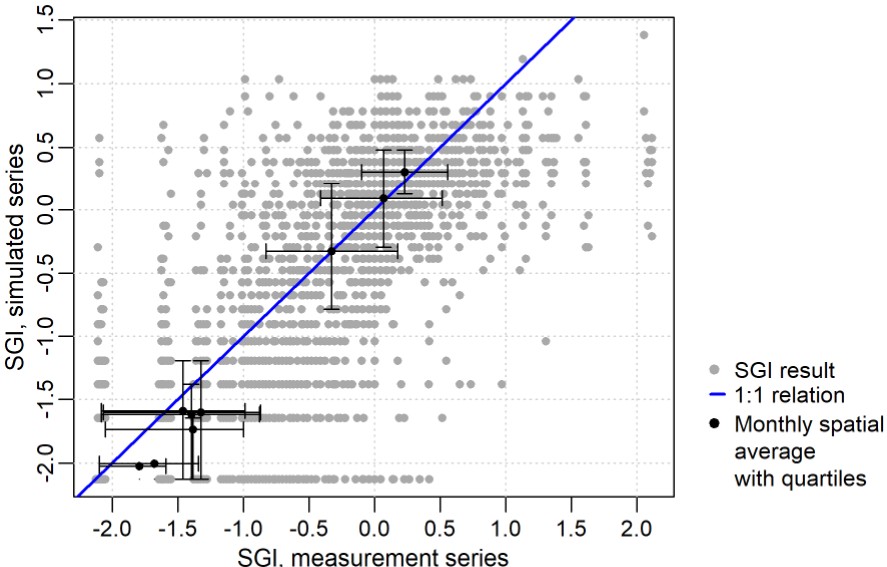

**Figure 8: Comparison of SGI values over April-December 2018 based on cleaned long measurement series (x axis) and based on simulated series (y axis) for all location-month combinations (grey dots). Black dots show the average over all locations for each month, with quartiles (spatial variation).**

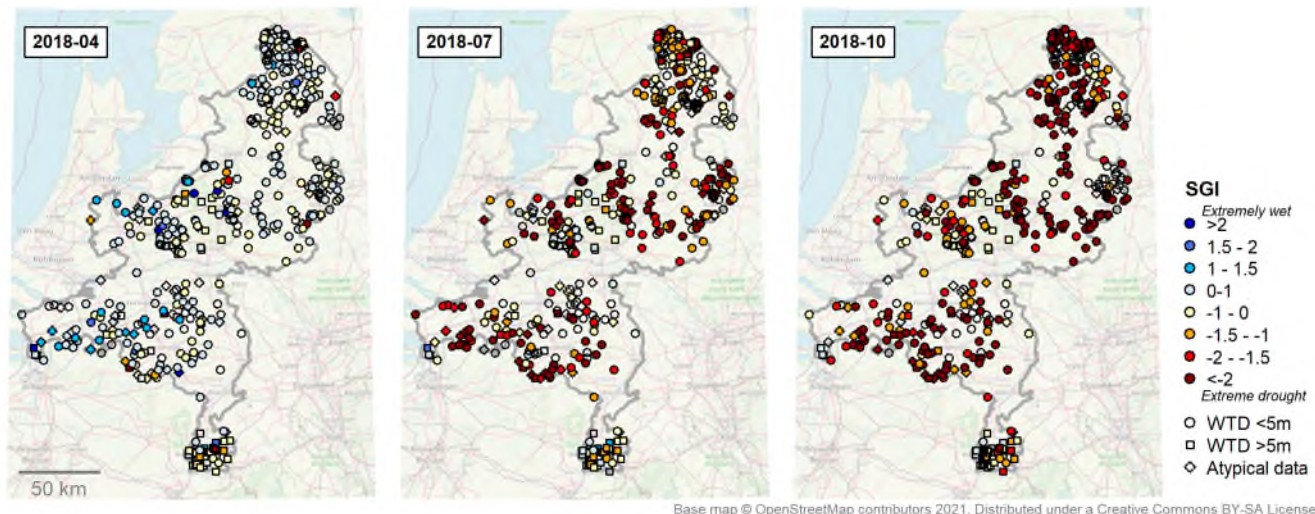


**Figure 9: SGI for three months in 2018 based directly on cleaned long measurement series. Empty symbols indicate insufficient data to enable drought index quantification.**

## 5 Discussion

### 5.1 Usefulness of TSM data preparation for drought analysis

The performed study aimed to evaluate the usefulness of time series modelling-based data processing methods for groundwater drought studies. The application of a TSM-based method to the 2018-2019 drought in the Netherlands has provided new insights into how TSM methods can be used for data validation, how reliable they are for the quantification of extreme groundwater drought situations, and how they can contribute to regional groundwater drought assessments.

#### 5.1.1 Validation methods for groundwater data

A validation method was applied that combines basic error tests with time series modelling-based identification of irregularities, while treating short-term outliers and long-term atypical behaviour separately. Pre-analysis outlier cleaning was found to improve the useability of series for the TSM simulations as well as the identification of long-term series behaviour. Outlier cleaning is therefore likely to improve results compared to performing a more limited validation, or identifying impacted series only after simulation. The validation method performed reasonably with regard to outlier removal, but not

optimally (Table 3, Table A3). The time-series based outlier cleaning appeared more effective in comparison to basic cleaning methods, although the latter can be valuable as a computationally cheap cleaning step for large data sets.

The TSM-based validation also appeared suitable for identifying long-term atypical behaviour in the head measurement series. However, the thresholds for separating series in different reliability classes (here 'discard' and 'atypical') are likely to be dependent on the data used and the aim of the study, and are difficult to set objectively. The current study aimed to provide a

spatially detailed regional drought assessment, covering the range of site conditions; this requires retaining as many series as possible, while removing the most unreliable and disturbed series and identifying series with milder potentially unreliable behaviour. The chosen parameters were found reasonable to reach this separation (see Fig. A5, A6) but setting objective parameter values would require comparison with a large data set with detailed information on the sources of variations and errors, which was not available in this study.

In this study, we could not explicitly separate erroneous and real atypical patterns. Many of the long-term non-weather influences on the groundwater, such as structural abstractions and land use, are likely to be included implicitly in the simulations, as the time series models will model any external influence that correlates with weather or modifies the recharge response. Still, the validation is likely to retain some series with erroneous patterns and discard some real behaviour. The separation of errors could be improved by including additional driving factors in the time series modelling, such as surface

water fluctuations (Bakker and Schaars, 2019; Van Loon et al., 2016; Zaadnoordijk et al., 2019); however, this requires substantially more data and modelling effort. Another potential approach is to make use of the spatial coherence of groundwater behaviour to separate errors. For example, Lehr and Lischeid (2020) and Marchant and Bloomfield (2018) used the spatial coherence of observed groundwater patterns as an indication of their reliability, and to relate clusters of similar series to some external (human) impact. It would be valuable to explore such extensions of TSM to improve validation methods for

groundwater drought analysis. This will allow for a better understanding of the different natural ánd human drivers of drought development.

Although the validation-simulation method generally performed well for the given data, it was less suitable for locations with deep groundwater tables (here > 5 m), dominated by multi-year head fluctuations. Here, model fit was often low, leading to poor outlier identification and models being discarded for a large proportion of these locations (66 % discarded vs 27 % for shallower locations). This issue was also found by Marchant and Bloomfield (2018) and Zaadnoordijk et al. (2019) for locations with thick unsaturated zones. To enable TSM simulation in such cases, measurement series are needed that are substantially longer than the minimum of 10 years used here to include several response cycles of the groundwater system.

### 5.1.2 Reliability of TSM groundwater level simulations during extreme drought

It was found that impulse-response function-based time series models on a general level produced reliable groundwater head simulations. They described most of the groundwater head series very closely, with a low overall bias (Table 5). The comparison of simulation- and measurement-based SGI values (Fig. 8) showed a good correlation (Spearman's $\rho = 0.8$). Also, as discussed further below, the general patterns shown by the drought index maps match well with the experience of water managers during the 2018-2019 drought.

However, the model simulations also showed important deviations, especially at local scales and during the most severe drought periods. The simulation RMSE and mean errors amounted locally to high levels (Table 5), especially during more extreme conditions. Also expressed as fraction of the WTD large errors occurred (Table 5). This means that application of the current TSM method could lead to misinterpretations of drought impacts at local scales, for example on the survival of marsh vegetation or groundwater-dependent streams (Bartholomeus et al., 2012; Witte et al., 2020b). Generally, extreme conditions were underestimated somewhat by the time series models. This overall underestimation of extreme conditions was also found by Mackay et al. (2015) with a process-based groundwater model. Interestingly, the simulations for summer 2018 showed an overestimation of drought conditions, where the simulations appeared to miss the variability towards less extreme drought that is visible in the measurements (Figs. 8, 9).

The model deviations during drought may be due to external influences not accounted for by the model, such as surface water influence and local irrigation, both of which are widespread in the study area. Indeed, local influences such as external water supply were found to alleviate groundwater drought locally in 2018 (Van den Eertwegh et al., 2019). In addition, the models may have underestimated the severity of earlier droughts; this would inflate the extremeness of the 2018 drought. Similar bias issues can be seen in Hellwig et al. (2020), who analysed groundwater drought in Germany using a physically-based model. Their simulations overestimated drought severity during a drought in 1973, but not in 2003, confirming that bias can differ between individual drought events.

In addition, the deviations may be related to the model setup itself. As noted before, we have here worked with a simple linear recharge model. Although this has been shown to suffice in many situations (e.g. Zaadnoordijk et al., 2019; von Asmuth et al., 2012), there are also various cases where linear recharge models were shown to be invalid (Bakker and Schaars, 2019;

Collenteur et al., 2021). Nonlinear or threshold responses in the soil-groundwater system may be especially important during extreme conditions, as evapotranspiration and deep percolation may become limited by low soil moisture or drainage becomes

disconnected (Bakker and Schaars, 2019; Aulenbach and Peters, 2018; Peterson and Western, 2014). Indeed, several studies have shown that nonlinear recharge representations can improve model performance during (extreme) drought conditions (Berendrecht et al., 2006; Peterson and Western, 2014; Collenteur et al., 2021). The overestimation of drought severity in 2018 found here (Fig. 8) may, among other factors, be related to a lack of evapotranspiration limitation in the models (Collenteur et al., 2021). The effect of extreme drought on evapotranspiration and flowpaths can be complex and spatially variable (Teuling

et al., 2013; Avanzi et al., 2020). Further exploring the value of nonlinear time series models for groundwater drought analyses in different situations is therefore an important topic for further research.

The reliability of groundwater simulations during extreme drought was found to be sensitive to the used calibration period, with reliability decreasing when the calibration period lacked similarly extreme conditions. This is an important aspect for the application of TSM for groundwater level nowcasting for real-time drought assessment. However, the difference in simulation

performance compared to simulations that did include the 2018 drought in their calibration was relatively small (Table 5-6). This suggests that the type of impulse-response time series models used in this study are relatively robust for series lengthening and nowcasting, also in drought conditions. This is also shown by the lengthened series over 2019-2020, which matched well with general observations provided by other studies and reports (LCW, 2020; Van den Eertwegh et al., 2019). Still, if water managers are to be provided structurally with up-to-date information on regional groundwater drought, any models have to be

re-calibrated frequently, so that reliable, recent measurement data remains important.

Despite the potential loss of information caused by TSM-based data preparation, consisting of processes and external influences not included in the models, it also provides important gains. Given the spatial variability in hydrological drought dynamics and the need for long data series for drought identification, TSM methods can be especially useful in drought studies to increase the amount of data available without a need for additional site information. In this study, it enabled a regional image

of drought development (Figs. 5, 6) that is far more detailed than that obtained by using direct measurement series only (Fig. 9). In addition, the model parameters gave additional insight in groundwater behaviour, for example through the response time (Bakker and Schaars, 2019; Collenteur et al., 2019). TSM data preparation, especially if further developed, can therefore likely be useful in many regional groundwater drought studies. However, in some cases direct use of data, inclusion of more external drivers or (a combination with) more physically-based model methods will be more suitable (e.g. Bakker and Schaars, 2019)

as in this study may have been the case for more local scales. The application of TSM methods therefore remains dependent on the aim and scale of a drought case study.

## 5.2 Analysis of the 2018-2019 groundwater drought in the Netherlands

### 5.2.1 Spatiotemporal development of groundwater drought in 2018-2019

The regional-scale analysis of the 2018-2019 groundwater drought in the southeastern Netherlands provided a new, detailed image of the drought development in time and space. The analysis showed that extreme groundwater drought was widespread over the study region in 2018 and 2019, breaking 30-year records for the summer and autumn months almost everywhere (Fig. 4). However, the timing and duration of the drought varied strongly in space. In the western parts of the study area drought was terminated before the end of 2018, while the higher-lying areas reached drought conditions only by 2019 (Fig. 5-6). This image corresponds well with how the drought was generally experienced by water managers. The Dutch Commission on Water allocation (LCW), which regularly surveys the drought status in the Netherlands based on input from water managers, reported widespread exceptional drops in groundwater levels, especially in the higher-lying sandy areas of the country, combined with a relatively fast recovery in the west (LCW, 2020). Also the severe damage observed in groundwater-dependent ecosystems in the southeastern Netherlands (Witte et al., 2020a) and widespread drying of groundwater-fed streams (Van den Eertwegh et al., 2019) match the assessment of an extreme groundwater situation.

The time series models made it possible to extend the available groundwater head measurement series beyond 2018 and obtain an estimate of drought recovery dynamics over 2019-2020. This showed that despite near-normal to high rainfall in the winters of 2018-19 and 2019-20 (Fig. 3) groundwater levels were restored only locally, and severe groundwater drought continued into 2019 or even 2020 over large parts of the study area (Fig. 6). These findings are consistent with general reports of the situation by local water managers (LCW, 2020; Van den Eertwegh et al., 2019). The results confirm that the drought should be viewed as a multi-year event rather than a single-year summer drought. The multi-year character increases the risk of lasting negative effects, especially in natural ecosystems. In addition, it stresses the importance of winter season groundwater management and water retention as determining factors in drought development. Both these issues have already become apparent in the Netherlands after 2018 (De Lenne and Worm, 2020; Witte et al., 2020a; Witte et al., 2020b).

### 5.2.2 Driving factors of spatial drought distribution

The current study did not aim to fully quantify the driving factors of the spatial variations in drought dynamics. However, the results suggest that the variation in drought severity, timing and duration was governed mainly by the spatial distribution in rainfall and the geological-topographic setting. The influence of spatial variations in weather was visible in late 2018 and 2019. In this period, a gradient in meteorological drought towards the east caused groundwater drought to be concentrated in this area. The effect of the geological-topographic setting is visible for the higher elevated parts of the study area, especially the glacial sand ridges and the limestone-loess hills, which clearly had a later, longer-lasting drought response than lower-lying areas (Figs. 5-6). This pattern is also visible in the groundwater response times (Fig. 7). The slow drought response in the higher elevated areas is likely explained by a thick unsaturated zone and low drainage density, while the fast recovery of the low-lying western parts may have been aided by thinner unsaturated zones and possibly some surface water influence. These

factors have been found by other studies to influence drought behaviour (Bloomfield et al., 2015; Hellwig et al., 2020; Peters
et al., 2006; Van Loon and Laaha, 2015; Kumar et al., 2016). In addition, the topographic-geological setting in the study area
correlates with variations in land use, soil type and aquifer characteristics; these factors may have played an additional role.
The dominant role of landscape position in shaping drought development calls for locally adapted, but also regionally
coordinated mitigation strategies. The response times obtained from the time series analysis form a useful first indicator of the
landscape characteristics that control the propagation of meteorological drought to groundwater drought. The response times
found here are somewhat shorter than the drought response times reported by Van Loon et al. (2017) for part of the eastern
Netherlands. However, they are very similar to those found by Zaadnoordijk et al. (2019), who performed time series analysis
on groundwater series from the whole Netherlands. As they used different data and different model quality criteria, the
similarity is reassuring and points to the stability of the time series models.

**6 Conclusions**

The performed study aimed to evaluate the usefulness of time series modelling-based data processing methods for regional-
scale groundwater drought assessment. A TSM-based method was set up for data validation and drought quantification and
applied to the regional 2018-2019 groundwater drought in the southeastern Netherlands to test its usefulness and reliability.
Automated TSM-based data validation was found able to improve the quality of input data. However, optimal validation
parameters are likely to be context-dependent. In addition, improvements in the validation method are desired, especially in
the separation of real and erroneous head disturbances. The simulated groundwater head series were generally found to be
reliable; however, it was shown that the use of time series simulations may bias drought estimations and underestimate spatial
variability, producing large errors at a local scale. Still, the use of time series model simulations in drought analysis provides
large advantages, as it enables a spatially detailed record of drought development that may be impossible to obtain with direct
measurement series only.
The drought analysis for the southeastern Netherlands provided a complete, detailed image of the development of the 2018-
2019 groundwater drought in time and space. The findings confirm that the meteorological drought in 2018 caused extreme
groundwater drought throughout the southeastern Netherlands, starting in late spring and peaking in October-November of
that year. The timing of drought onset and the duration of drought varied strongly in space. Drought development appeared to
be governed dominantly by the spatial distribution of rainfall and the geological-topographic setting. In much of the area, the
drought continued as a multi-year drought into 2019 and 2020, especially in the eastern and higher elevated regions.
Taken together, we conclude that time series modelling forms a useful tool to obtain a fast, detailed and up-to-date image of
drought development; however, for a proper understanding of the different driving factors of drought the availability of recent
and consistent monitoring data remains crucial.

## Appendix A

 **A1 Time series model setup**

Groundwater head modelling was done by impulse-response transfer function-noise models using the Pastas package in Python (Python 3.6, Pastas 0.13). The model setup used in this study largely follows Collenteur et al. (2019). The code for building and simulating the models, as applied in the validation and simulation steps in this study, is given at the end of this section.

The working of impulse-response type transfer function-noise models for groundwater series is explained in von Asmuth (2002) and Collenteur et al. (2019). In principle, the method models groundwater heads as:

$$h(t) = \sum_{m=1}^{M} h_m(t) + d + r(t) \tag{A1}$$

where $h_m$ are the variations in heads caused by one or several stresses $m$; $d$ is the base level; and $r$ the residual at time $t$. Here only recharge was used as explanatory variable. Recharge is estimated as a linear combination of precipitation $P$ and reference evapotranspiration $ET_{ref}$:

$$R(t) = P(t) - f \cdot ET_{ref} \tag{A2}$$

With $f$ a model parameter. The response of groundwater to a recharge impulse is calculated with a scaled gamma function (option *ps.Gamma* in Pastas):

$$\theta(t) = A \frac{t^{n-1}}{a^n \Gamma(n)} e^{-t/a} \tag{A3}$$

With $\Gamma$ the Gamma function, $A$ a scale parameter and $a$ and $n$ shape parameters. The variation in groundwater heads over time $h_m$ is obtained by convolution of this impulse-response function with the recharge time series.

This gives four parameters to be fit for each individual location. Parameter $A$ represents the long-term response of the groundwater level to a constant recharge input of one unit, in this case 1 mm; $f$ is the influence of $ET_{ref}$ relative to precipitation; and $d$ is the groundwater base level. For purposes of parameter calibration, also an AR(1) noise model is fit to the residuals, giving the additional noise decay parameter $\alpha$. The default method for parameter optimisation was used, which minimises the sum of weighted squared noise by a least squares method (*ps.LeastSquares*). Table A1 gives the calibration settings for each of the parameters, as well as the range found in the raw groundwater head series in this study. Based on these ranges additional bounds were set for parameters $A$ and $f$ for the long-term behaviour classification (see section 3.2); series with unusual parameters beyond these bounds were classified as 'atypical' and potentially unreliable.

**Table A1: Parameter calibration settings: initial value and bounds during optimisation (default settings Pastas); the range of values found in this study when modelling the initial 2723 cleaned series; and reliability bounds used for the series classification.**

| | Initial value (Pastas) | Allowed range (Pastas) | Found values in this study (10th-90th percentile) | Reliability bounds for classification |
|---|---|---|---|---|

| | | | | |
|---|---|---|---|---|
| A | $1/\sigma_{recharge}$ | >0 | 0.10…1.2 | <1.5 |
| a | 10 | | 72…473 | |
| n | 1 | | 0.61…1.9 | |
| f | -1 | -2…0 | -1.4…-0.44 | -1.95…-0.05 |
| d | Mean of series | | 0.95..41 | |
| α | 15 | | 17…636 | |

Except for the simulations used to test the calibration period effect, the full period of available data between 1990 and 2019 was used for calibration. Calibration was done at a weekly timestep (recharge and observation series subsampled to a weekly frequency before optimisation) to allow a more even spread of optimisation points over time. Simulation of series was done at 610 a daily timestep from 1990-01-01 to 2020-05-31.

```
"""
Basic time series model function for groundwater heads.
Based on Collenteur et al. (2019) doi: 10.1111/gwat.12925
Used in groundwater drought in the southern Netherlands project
Esther Brakkee, spring 2020
Applied with Pastas 0.13 and Python 3.6
"""
import pandas as pd
import Pastas as ps

def PastasModelParsSim (heads,rain,evap,calibstart="1990-01-01",
                        calibend="2019-12-31",simstart="1990-01-01",
                        simend="2020-05-31",method="Linear"):
    '''
    function to make a basic impulse-response time series model
    INPUT:
        -head dataframe with 'DateTime' in '%d-%m-%Y %H:%M' and 'Head' in m+MSL
        -daily-step series for rain and reference evapotranspiration, with
       'DateTime' in '%d-%m-%Y %H:%M' and 'Precip'/'Evap' in mm/d
    PARAMETERS:
        -calibstart, calibend: start and end of calibration period in
        '%Y-%m-%d'
        -simstart, simend: same for simulation period
        -ONLY LINEAR MODELLING FOR NOW
    OUTPUT:
        dictionary with simulated series, model parameters, fit criteria,
        residuals and noise
    '''
    #create model
    ml = ps.Model(heads, name='Model')
    if method=="Linear":
        rm = ps.RechargeModel(rain, evap, rfunc=ps.Gamma)#linear is default
    ml.add_stressmodel(rm)
    #solve model with weekly timestep
    ml.solve(freq="7D",tmin=calibstart,tmax=calibend,report=False)
    #get output
    res=ml.residuals()
    pars=ml.get_parameters() #A,n,a,f,d,noise-alpha
    evp=ml.stats.evp()
    noise=ml.noise()
    sim=ml.simulate(tmin=simstart, tmax=simend, freq="D")
    #all outcomes into a dictionary
    outcomes={"EVP":evp,"parameters":pars,"residuals":res,"method":method,
              "sim":sim,"noise":noise}
    return outcomes
```

## A2 Validation parameter sensitivity

### A2.1 Parameter sets

The validation routine was run with 21 different parameter sets (table A2). In set 2-11, the parameters are varied individually;
in set 12-19, combinations of outlier cleaning and long-term deviation identification parameters are tested; and in set 20 and
21 versions are tested with only TSM-based outlier cleaning (no basic cleaning step) and no outlier cleaning at all.

**Table A2: Used parameter sets for the sensitivity test. See table 1 for explanation of the parameters.**

| Nr | Name | Meta Check | $F_{range}$ | $TH_{red}$ | $n_{SD}$ | $n_{iter}$ | $TH_{EVP}$ | $TH_{r2}$ |
|---|---|---|---|---|---|---|---|---|
| *1* | *Standard* | *Yes* | *0.2* | *0.5* | *4* | *2* | *60* | *0.15* |
| 2 | FarOutliersConservative | Yes | 0.1 | 0.7 | 4 | 2 | 60 | 0.15 |
| 3 | FarOutliersRigorous | Yes | 0.2 | 0.4 | 4 | 2 | 60 | 0.15 |
| 4 | OutliersConservative | Yes | 0.2 | 0.5 | 6 | 2 | 60 | 0.15 |
| 5 | OutliersRigorous | Yes | 0.2 | 0.5 | 2 | 2 | 60 | 0.15 |
| 6 | IterationsConservative | Yes | 0.2 | 0.5 | 4 | 1 | 60 | 0.15 |
| 7 | IterationsRigorous | Yes | 0.2 | 0.5 | 4 | 5 | 60 | 0.15 |
| 8 | EVPConservative | Yes | 0.2 | 0.5 | 4 | 2 | 40 | 0.15 |
| 9 | EVPRigorous | Yes | 0.2 | 0.5 | 4 | 2 | 80 | 0.15 |
| 10 | TrendConservative | Yes | 0.2 | 0.5 | 4 | 2 | 60 | 0.4 |
| 11 | TrendRigorous | Yes | 0.2 | 0.5 | 4 | 2 | 60 | 0.05 |
| 12 | OutliersConservative_EVPConservative | Yes | 0.1 | 0.7 | 6 | 1 | 40 | 0.15 |
| 13 | OutliersConservative_EVPRigorous | Yes | 0.1 | 0.7 | 6 | 1 | 80 | 0.15 |
| 14 | OutliersConservative_TrendConservative | Yes | 0.1 | 0.7 | 6 | 1 | 60 | 0.4 |
| 15 | OutliersConservative_TrendRigorous | Yes | 0.1 | 0.7 | 6 | 1 | 60 | 0.05 |
| 16 | OutliersRigorous_EVPConservative | Yes | 0.2 | 0.4 | 2 | 5 | 40 | 0.15 |
| 17 | OutliersRigorous_EVPRigorous | Yes | 0.2 | 0.4 | 2 | 5 | 80 | 0.15 |
| 18 | OutliersRigorous_TrendConservative | Yes | 0.2 | 0.4 | 2 | 5 | 60 | 0.4 |
| 19 | OutliersRigorous_TrendRigorous | Yes | 0.2 | 0.4 | 2 | 5 | 60 | 0.05 |
| 20 | Standard_TSMcleaningOnly | No | 0 | 1 | 4 | 2 | 60 | 0.15 |
| 21 | Standard_NoOutlierCleaning | No | 0 | 1 | 100 | 0 | 60 | 0.15 |

## A2.2 Test results

### Outliers

Table A3 gives the full outlier cleaning performance results for all parameter sets. Cleaning of outliers in general was able to increase the EVP of series TSM models from 60 % to 62 % on average (set 1 vs 21). Also, applying a cleaning step allows for more series to be retained. The basic, range-based outlier step adds relatively little to the cleaning quality compared to the time series model-based cleaning (set 20 vs 1). The range-based cleaning step appeared useful in a small number of cases where it
improved the quality of the TSM cleaning; in addition, it is a computationally cheap step and was therefore retained in the validation.

As expected from the minor effect of the far outlier cleaning, the thresholds of this step have little effect on the validation performance (set 2 and 3). Changing the TSM-based outlier cleaning thresholds does result in large effects: lowering the SD

threshold to 2·SD (set 5) causes the mean EVP (logically) to increase substantially, but also causes removal of many data
points that would visually not be identified as errors (many false positives). Applying a more conservative TSM outlier cleaning
with 6·SD (set 4) has the reverse effect, with many outliers not identified and more series being discarded. The number of
iterations applied in the TSM outlier cleaning only slightly affected the resulting EVP of the series models. Apparently the
first cycle already cleans the main outliers in most cases.

**Table A3: Validation performance with regard to outliers. Left columns: number of series (of *n*=180) classified in each category.**
**TP=outliers identified in manual and automatic validation; TN=outliers not identified in either manual or automatic validation; FP=outliers identified in automatic validation but not in manual validation; FN=outliers identified in manual but not in automated validation. Last column: percentage of series with outliers correctly cleaned.**

| Set | Name | Missing Data | Outliers TP | Outliers TN | Outliers FP | Outliers FN | Mean EVP | Outliers Good [%] |
|-----|------|--------------|-------------|-------------|-------------|-------------|----------|-------------------|
| 1 | Standard | 19 | 71 | 56 | 26 | 8 | 61.9 | **79** |
| 2 | FarOutliersConservative | 19 | 72 | 56 | 26 | 7 | 62 | **80** |
| 3 | FarOutliersRigorous | 19 | 71 | 56 | 26 | 8 | 61.9 | **79** |
| 4 | OutliersConservative | 19 | 36 | 92 | 5 | 28 | 60.6 | **80** |
| 5 | OutliersRigorous | 19 | 96 | 0 | 64 | 1 | 68.8 | **60** |
| 6 | IterationsConservative | 19 | 71 | 56 | 26 | 8 | 61.7 | **79** |
| 7 | IterationsRigorous | 19 | 71 | 56 | 26 | 8 | 62.2 | **79** |
| 8 | EVPConservative | 19 | 71 | 56 | 26 | 8 | 61.9 | **79** |
| 9 | EVPRigorous | 19 | 71 | 56 | 26 | 8 | 61.9 | **79** |
| 10 | TrendConservative | 19 | 71 | 56 | 26 | 8 | 61.9 | **79** |
| 11 | TrendRigorous | 19 | 71 | 56 | 26 | 8 | 61.9 | **79** |
| 12 | OutliersConservative_ EVPConservative | 19 | 37 | 92 | 5 | 27 | 60.5 | **80** |
| 13 | OutliersConservative_ EVPRigorous | 19 | 37 | 92 | 5 | 27 | 60.5 | **80** |
| 14 | OutliersConservative_ TrendConservative | 19 | 37 | 92 | 5 | 27 | 60.5 | **80** |
| 15 | OutliersConservative_ TrendRigorous | 19 | 37 | 92 | 5 | 27 | 60.5 | **80** |
| 16 | OutliersRigorous_ EVPConservative | 22 | 93 | 0 | 64 | 1 | 71.1 | **59** |
| 17 | OutliersRigorous_ EVPRigorous | 22 | 93 | 0 | 64 | 1 | 71.1 | **59** |
| 18 | OutliersRigorous_ TrendConservative | 22 | 93 | 0 | 64 | 1 | 71.1 | **59** |
| 19 | OutliersRigorous_ TrendRigorous | 22 | 93 | 0 | 64 | 1 | 71.1 | **59** |
| 20 | Standard_ TSMcleaningOnly | 19 | 73 | 54 | 26 | 8 | 62.1 | **79** |
| 21 | Standard_ NoOutlierCleaning | 19 | 0 | 110 | 0 | 51 | 59.9 | **68** |

**Serious long-term deviations: discarding of series**

Table A4 shows the validation performance with regard to serious long-term deviations in the series. Changing the EVP threshold for discarding series logically has a strong effect on the number of discarded series (set 8 and 9). Taking a conservative low EVP threshold appears to give a good performance on the strong deviation identification (set 8 and 16), but the number of false negatives, leading to potentially erroneous outcomes, is high. The outlier cleaning also affects the identification of long-term deviations. More rigorous outlier cleaning, especially by reducing the SD threshold, led to

discarding a smaller number of series and slightly better performance on the long-term deviation identification, with conservative outlier cleaning having the reverse effect (set 2-5). However, the changes are small.

**Table A4: Validation performance with regard to strong long-term deviation. Left columns: number of series (of *n*=180) classified in each category. TP=discarded in manual and automatic validation; TN=not discarded in either manual or automatic validation; FP=discarded in automatic validation but not in manual validation; FN=discarded in manual but not in automated validation. Excl**

**deep: false positives excluding deep-GWL series. Discard number: number of series discarded of *n*=180. Last column: percentage of series with long-term strong deviation correctly identified.**

| Set | Name | Discard TP | Discard TN | Discard FP | Discard FP excl deep | Discard FN | Discard number | Discard Good [%] |
|---|---|---|---|---|---|---|---|---|
| 1 | Standard | 32 | 103 | 26 | 22 | 0 | 58 | **84** |
| 2 | FarOutliersConservative | 32 | 103 | 26 | 22 | 0 | 58 | **84** |
| 3 | FarOutliersRigorous | 32 | 103 | 26 | 22 | 0 | 58 | **84** |
| 4 | OutliersConservative | 33 | 97 | 31 | 27 | 0 | 64 | **81** |
| 5 | OutliersRigorous | 24 | 115 | 17 | 13 | 5 | 41 | **86** |
| 6 | IterationsConservative | 32 | 100 | 29 | 25 | 0 | 61 | **82** |
| 7 | IterationsRigorous | 32 | 103 | 26 | 22 | 0 | 58 | **84** |
| 8 | EVPConservative | 17 | 121 | 12 | 8 | 11 | 29 | **86** |
| 9 | EVPRigorous | 34 | 42 | 85 | 73 | 0 | 119 | **47** |
| 10 | TrendConservative | 32 | 103 | 26 | 22 | 0 | 58 | **84** |
| 11 | TrendRigorous | 32 | 103 | 26 | 22 | 0 | 58 | **84** |
| 12 | OutliersConservative_ EVPConservative | 17 | 121 | 12 | 8 | 11 | 29 | **86** |
| 13 | OutliersConservative_ EVPRigorous | 34 | 39 | 88 | 74 | 0 | 122 | **45** |
| 14 | OutliersConservative_ TrendConservative | 33 | 95 | 33 | 29 | 0 | 66 | **80** |
| 15 | OutliersConservative_ TrendRigorous | 33 | 95 | 33 | 29 | 0 | 66 | **80** |
| 16 | OutliersRigorous_ EVPConservative | 12 | 129 | 3 | 1 | 14 | 15 | **89** |
| 17 | OutliersRigorous_ EVPRigorous | 32 | 76 | 50 | 41 | 0 | 82 | **68** |
| 18 | OutliersRigorous_ TrendConservative | 21 | 116 | 15 | 11 | 6 | 36 | **87** |
| 19 | OutliersRigorous_ TrendRigorous | 21 | 116 | 15 | 11 | 6 | 36 | **87** |
| 20 | Standard_ TSMcleaningOnly | 32 | 104 | 25 | 21 | 0 | 57 | **84** |

| 21 | Standard_<br>NoOutlierCleaning | 33 | 97 | 31 | 27 | 0 | 64 | **81** |
|----|----|----|----|----|----|----|----|----|

**Mild long-term deviations: series to be marked as atypical**

The fraction of series marked as atypical in the data is relatively small across the different parameter sets. The used $r^2$ threshold
to identify trends affects this number. A high threshold of 0.4 results in very few atypical series being identified; a low threshold results in many false positives, as weak trends were present in a large proportion of the series. The identification of atypical series is also affected by the outliers cleaning parameters. More rigorous outlier cleaning (set 5) causes more series to be marked as atypical. Also the EVP threshold to discard series affects the number of series marked as atypical. With a high EVP threshold discards many mildly atypical series are discarded by an insufficient EVP. A low EVP threshold brings more series
into the 'atypical' category.

**Table A5: Validation performance with regard to mild long-term deviations. Left columns: number of series (of *n*=180) classified in each category. TP=marked as atypical in both manual and automatic validation; TN=not marked as atypical in either manual or automatic validation; FP=marked as atypical in automatic validation but not in manual validation; FN=marked as atypical in manual but not in automated validation. Excl deep: false positives excluding deep-GWL series. Atyp number: number of series**
**marked atypical of n=180. Last columns: percentage of series with mild long-term deviations correctly identified.**

| Set | Name | AtypTP | Atyp TN | Atyp FP | Atyp FP excl deep | Atyp FN | Atyp number | Atyp Good [%] |
|----|----|----|----|----|----|----|----|----|
| 1 | Standard | 7 | 131 | 11 | 7 | 12 | 18 | 86 |
| 2 | FarOutliersConservative | 7 | 131 | 11 | 7 | 12 | 18 | 86 |
| 3 | FarOutliersRigorous | 7 | 131 | 11 | 7 | 12 | 18 | 86 |
| 4 | OutliersConservative | 7 | 132 | 10 | 6 | 12 | 17 | 86 |
| 5 | OutliersRigorous | 9 | 123 | 19 | 14 | 10 | 28 | 82 |
| 6 | IterationsConservative | 7 | 131 | 11 | 7 | 12 | 18 | 86 |
| 7 | IterationsRigorous | 7 | 131 | 11 | 7 | 12 | 18 | 86 |
| 8 | EVPConservative | 10 | 119 | 23 | 18 | 9 | 33 | 80 |
| 9 | EVPRigorous | 0 | 142 | 4 | 2 | 15 | 4 | 88 |
| 10 | TrendConservative | 2 | 139 | 5 | 1 | 15 | 7 | 88 |
| 11 | TrendRigorous | 8 | 111 | 31 | 24 | 11 | 39 | 74 |
| 12 | OutliersConservative_<br>EVPConservative | 10 | 122 | 20 | 15 | 9 | 30 | 82 |
| 13 | OutliersConservative_<br>EVPRigorous | 0 | 142 | 4 | 2 | 15 | 4 | 88 |
| 14 | OutliersConservative_<br>TrendConservative | 2 | 139 | 5 | 1 | 15 | 7 | 88 |
| 15 | OutliersConservative_<br>TrendRigorous | 7 | 113 | 29 | 22 | 12 | 36 | 75 |
| 16 | OutliersRigorous_<br>EVPConservative | 13 | 106 | 32 | 24 | 7 | 45 | 75 |
| 17 | OutliersRigorous_<br>EVPRigorous | 2 | 135 | 6 | 4 | 15 | 8 | 87 |
| 18 | OutliersRigorous_<br>TrendConservative | 3 | 131 | 10 | 5 | 14 | 13 | 85 |

| 19 | OutliersRigorous_<br>TrendRigorous | 10 | 104 | 35 | 30 | 9 | 45 | 72 |
|----|------|----|-----|----|----|----|----|----|
| 20 | Standard_<br>TSMcleaningOnly | 8 | 131 | 11 | 7 | 11 | 19 | 86 |
| 21 | Standard_<br>NoOutlierCleaning | 8 | 134 | 8 | 4 | 11 | 16 | 88 |

## A3 Example figures

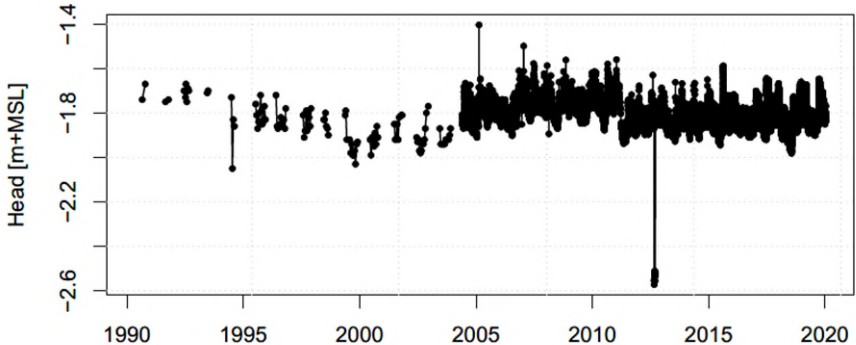

**Figure A1: Example of a series with a clearly undesired outlier. The isolated low points in 2013 are caused by temporary extraction to clean the well. The series also contains smaller outliers in e.g. 1994 and 2006.**

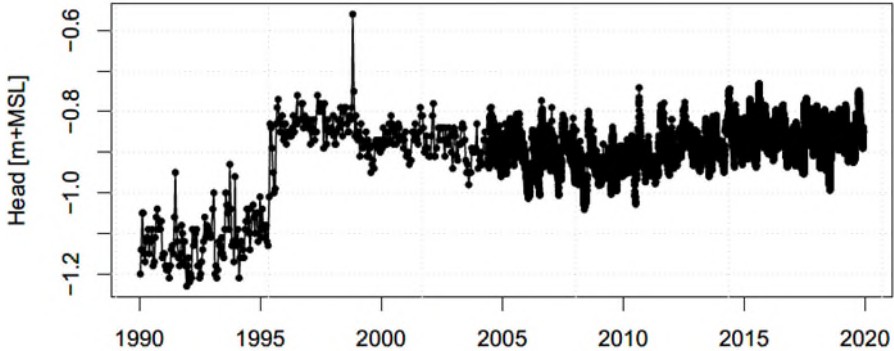

**Figure A2: Example of a series with a serious long-term deviation. The level shift around 1996 is caused by changes in water management in the area. Also an outlier in 1999.**

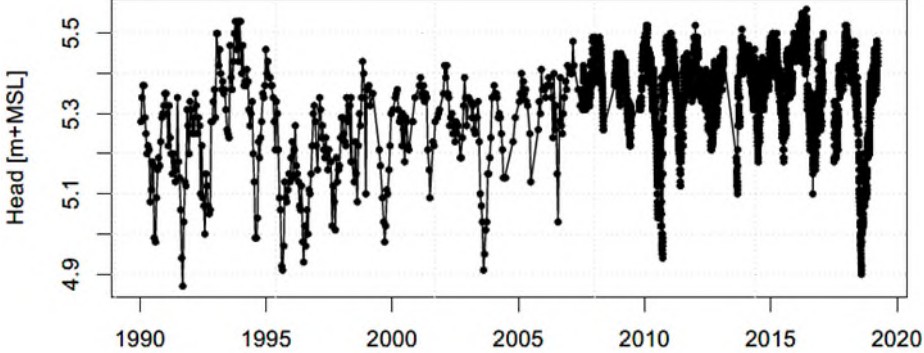

**Figure A3: Example of a series with potentially unreliable long-term behaviour. There seems to be a trend from 1996 and a stabilisation after 2006, but the variations could be caused by weather variations and do not give reason to discard the series.**

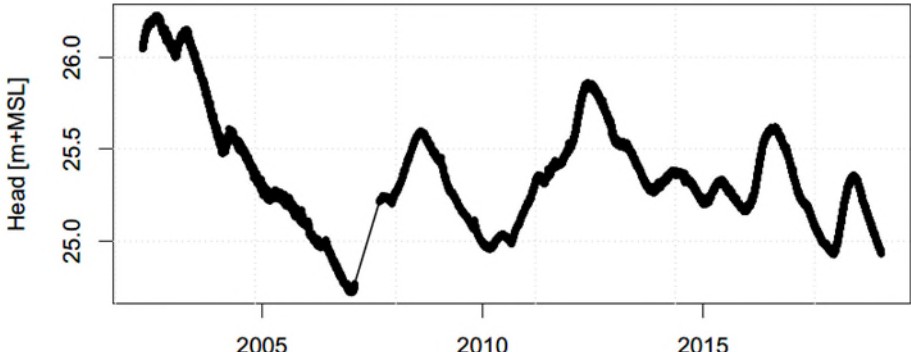

**Figure A4: Example of a series with a deep groundwater table and resulting multi-year variation. Series located on the Veluwe ice-pushed ridge, mean WTD 35 m.**

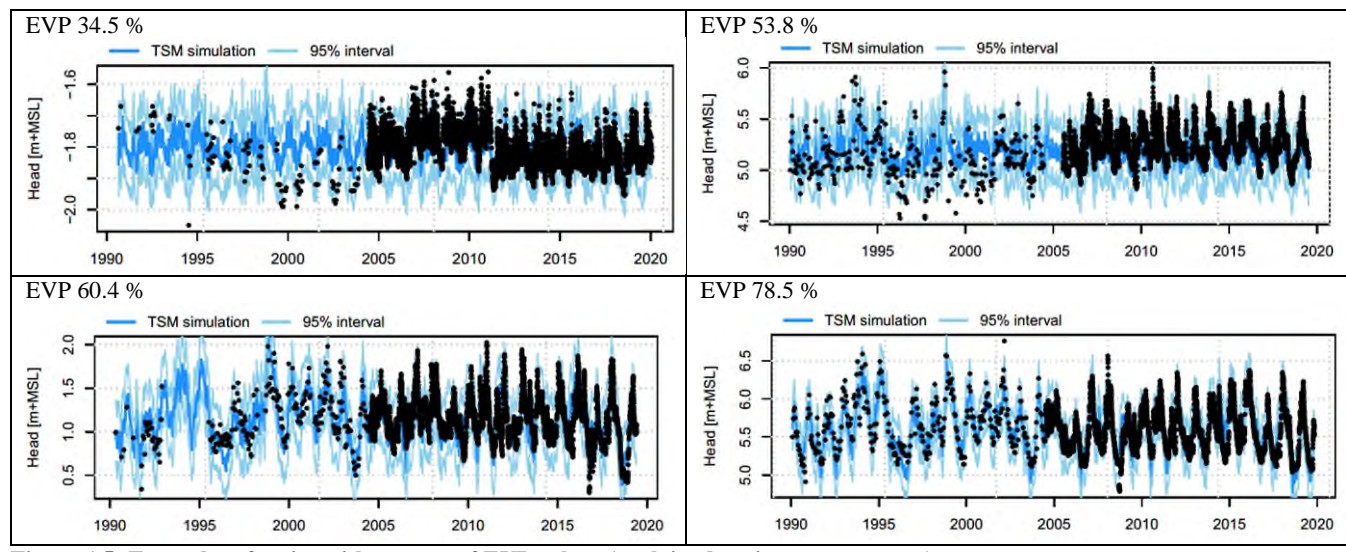

**Figure A5: Examples of series with a range of EVP values (explained variance percentage).**

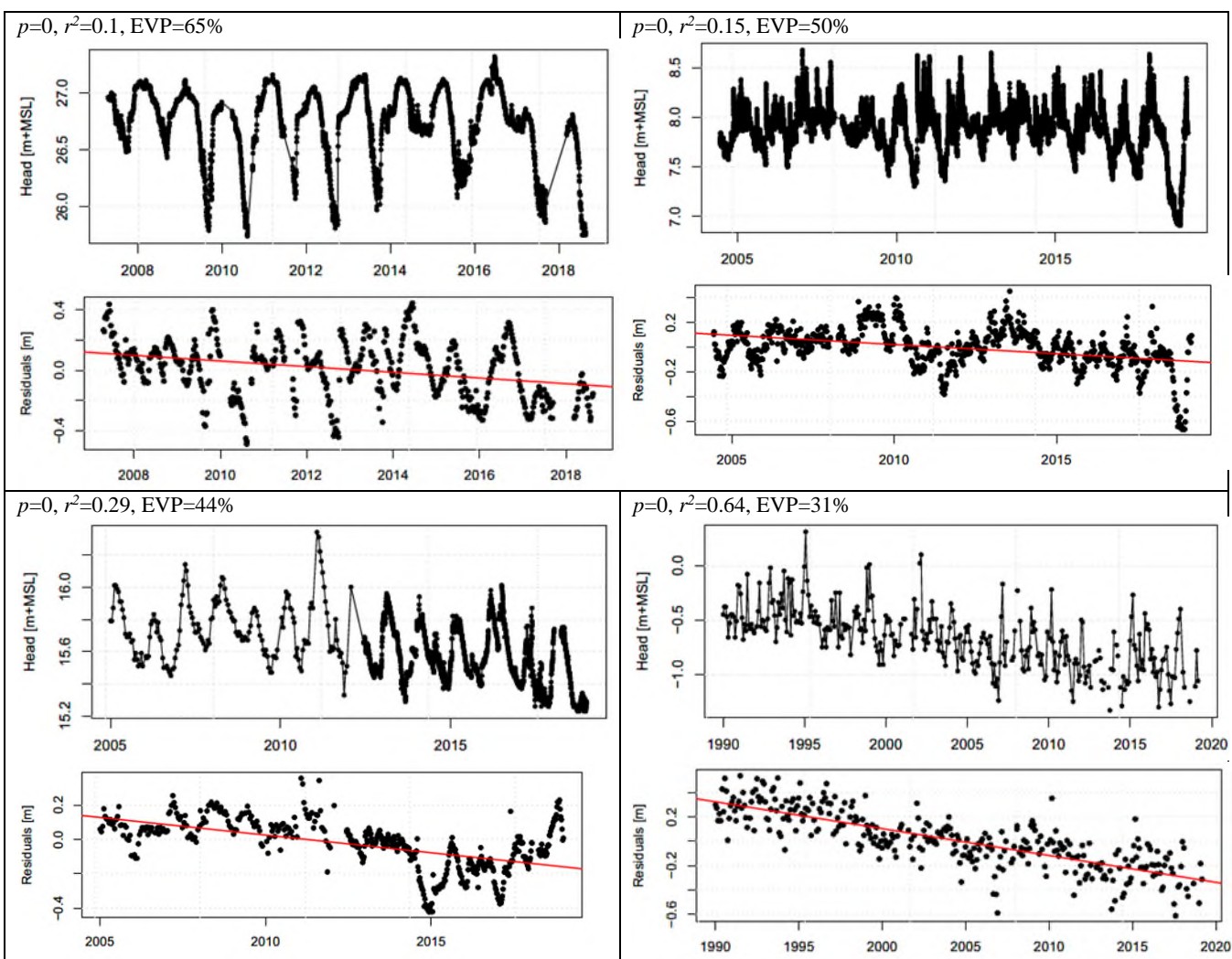

**Figure A6: Examples of series with a range of trend r² values in the model residuals. Both the series itself and the residuals with fitted trend are shown.**

## Code and data availability

All code for data processing, modelling and visualisation is available on request from the first author. Groundwater data are
freely available from the national groundwater database DINO ([www.dinoloket.nl](www.dinoloket.nl)). The used weather data can be obtained from the Royal Dutch Meteorological Institute (KNMI) via [http://projects.knmi.nl/klimatologie/daggegevens/selectie.cgi](http://projects.knmi.nl/klimatologie/daggegevens/selectie.cgi) or through a script via [https://www.knmi.nl/kennis-en-datacentrum/achtergrond/data-ophalen-vanuit-een-script](https://www.knmi.nl/kennis-en-datacentrum/achtergrond/data-ophalen-vanuit-een-script).

## Author contribution

EB, MvH and RB together designed the study. EB performed the study with advice from MvH and RB. The first article draft was written by EB and reviewed by all three authors.

## Competing interests

The authors declare that they have no conflict of interest.

## Acknowledgements

The authors thank Sharon Clevers for collecting and pre-processing an important part of the groundwater data.

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
