# Peer review of "Improved understanding of regional groundwater drought development through time series modelling: the 2018-2019 drought in the Netherlands"

_Hydrology and Earth System Sciences, 2021_

## Referee Comment (RC1)

**Review of: „Spatiotemporal development of the 2018-2019 groundwater drought in the Netherlands: a data-based approach" by Brakkee et al.**

The authors use an impulse-response time series modeling (TSM) approach to estimate groundwater heads over a larger region. By doing so, they were able to provide a much denser network of point-based groundwater drought estimates over a larger region, compared to using solely groundwater observation. The paper covers various interesting topics, including groundwater data quality assessments, groundwater time series grouping, groundwater drought analyses, and the „prediction" of groundwater drought. In addition, it suggests that some of the dominant controls on groundwater drought development are the spatial variability in precipitation and differences in geological-topographic conditions.

The TSM approach applied on the regional scale to increase the spatial density of groundwater drought information is novel (as far as I know) and I am impressed by the large amount of data that is collected and used for the analyses. That being said, there are quite some methodological aspects that should be better, explained, justified, or evaluated (listed below). In addition, all the proposed analyses could make for an interesting paper. However, at this moment, few of these analyses are thorough enough  to generate some more complete conclusions. Below, I provide various suggestions how the analyses could be elaborated, but the authors may have some different ideas. If the additional analyses make the paper to lengthy, the authors may choose to drop e.g. the prediction part. Besides the major remarks, I have various minor comments that should be addressed.

**Methodological aspects (listed in order of appearance).**

- Line 159: What is „short-term" (days, months, years?). And why are these not relevant. Please explain.
- Line 162: Why are long term disturbances relevant for drought studies? Please explain.
- Line 166: Why are time series with long-term changes less reliable for drought analyses (also given the prev. point). Or does this refer to their simulations?
- Line 170-71: I would prefer a bit more detailed description of the PASTAS approach here (I can read it from the mentioned publication, but a short summary of the approach and the used parameters would be easier for readers that do not want to go over all the details).
- Line 173: describe the used non-linear approach in more detail (or delete / move to discussion that you tested a non-linear approach and how).
- Line 176: why a cutoff at >20 cm. Is the reason that GW head can exceed surface level related to inaccuracies in the used elevation data?
- Line 177-178: the use of thresholds like 20% of data and 50% of the measurement ranges should be better justified or evaluated. Also, by focusing on both the wet and dry outliers at the same time, you might either: disregard some GW wells that are suitable for drought but have a lot of variance in the wetter domain or include some less suitable wells that have a high variance in the dry domain and a low variance in the wet domain. This should be discussed.
- Line 180: what is „basic model settings of PASTAS"
- Line 185: better explain A, f and d, and what these metrics mean.
- Line 202 (and other places): The term „aggregated" alone can mean various things. Be precise.
- Line 205: why randomly select 120 GW timeseries and then select 56 from those 120. Why not just select 56 randomly?
- Line 205-206: it would be helpful to show some exemplary timeseries with and without visual errors, preferably in the results (maybe in the supplement).
- Line 209-210: please provide abbreviations of true positive etc. here. Otherwise, few people will understand table 2.

- Line 218-219: please provide more detail about the PASTAS model as applied in this study here or in the previous section.
- Line 232: what are these „practical consequences"
- Line 238: which distribution is used for the SPEI?
- Line 240: why not simply the mean of all stations in a region?
- Line 250: „probably insufficient to do this reliably" either test this or delete.
- Line 256: the original SGI paper of Bloomfield and Marchant (2013) used 1 / (2n) and not 1 / n. Please explain why you differentiate.
- Table 1: values are not correct. E.g., if the lowest SGI is based on 1 / n (1 / 30), the lowest SGI value would be >-2.
- Line 269-272. When comparing observed vs simulated SGI, did you match their record lengths. For example, if the simulated SGI covers 1990-2019, but the observed SGI 1993-2019, did you recalculate the simulated SGI for 1993-2019?
- Line 294: did you combine simulated and observed data here?
- Line 360-361. This should be presented in the method section and more clearly explained.

**Results:**

- You used plenty of outlier removal approaches, but never really evaluated how these different approaches affected (improved) the simulations. For example, did removing the lowest and highest 20% of values for some wells result in better GW / SGI simulations. I strongly recommend repeating the evaluation presented in table 3 based on data that did not receive (various steps of) preprocessing. Here, it would be also interesting to look at model performance for the different subsets of wells (e.g. discarded, deep, atypical, typical). Finally, it might be interesting to investigate model performance as a function of the available record length. All in all, this would provide a more quantitative assessment of your data quality and outlier removal approach.
- You note that groundwater drought (propagation) characteristics appeared to be dominantly governed by the spatial distribution of rainfall and the geological-topographic setting (abstract). This statement would benefit from some more quantitative analyses. Would it somehow be possible to group wells based on common aquifer characteristics (if not, possibly just the wells in the highlighted areas vs. the wells in the other regions, Fig. 1) and investigate whether different subsets of wells respond differently to the drought of 2018-2020? Such analyses could be linked to the maps of figs. 5-7, possibly comparing the density functions of the different subgroups (like in Figure 4). Here, it might also be interesting to compare the different types of wells again.
- For the usefulness analyses, I strongly suggest evaluating the usefulness of your approach for the different groups of wells (typical, atypical, deep, discarded). Further, I would not purely focus on the drought of 2018-2019, but also on the entire period. In addition to deriving the correlation with SGI observations for all stations at once (Fig. 8), it would be interesting to see the (spread in) correlation of each individual well (and again see if these correlations vary among the different categories of wells).
- The prediction analysis is not really a prediction analyses but more a model evaluation (like you split a hydrological model experiment in a calibration and evaluation period). The section is generally ok, besides the last paragraph (406-411), which contains an interesting statement about the limited value of including the initial conditions for prediction. This statement is contrary to what I would expect from often highly autocorrelated groundwater timeseries. Therefore, it deserves more attention! I assume the stated values are averages? I think providing ranges here would be important, as I assume that there are definitely some wells that benefit from the inclusion of the initial condition. Grouping based on well type might work again (deep wells might possibly be predicted for longer lead-times). Possibly a Figure would be nice here

as well, with on the x-axis the lead time and on the y-axis the „improvement", showing both the average (e.g. bold line) as well as the ranges (background shade).

**Minor comments:**

- **Line 1:** „development": it is not just the development.
- **Line 1:** „2019" you include 2020.
- **Line 2:** „a data-based approach". This does not add a lot. I think all GW drought assessments are data based. You could be more specific with regard to the used method (or delete this part).
- **Line 5:** „northwestern". Parts of central Europe were affected as well.
- **Line 55:** „has so far mainly been studied from a meteorological perspective". I do not think that this is the case. The reference to Bakke includes hydrological observations and previously mentioned refs. study tree response to the drought of 2018. In addition, a reference of Brunner et al. (2019) would fit here nicely, as it was one of the first studies to investigate the 2018 drought (from different perspectives).
- **Line 66-67:** „few measurement wells" the mentioned study of Van Loon et al. (2017) used quite a lot of wells.
- **Line 66-67**: a reference to Kumar et al. (2016) might fit here.
- **Line 98-99:** please check if all mentioned references focus on GW forecasting. Also, I am a bit surprised that physical based models can only predict two months ahead, given the often strong autocorrelation in GW signals.
- **Line 119:** please briefly note why these regions are highlighted (higher vulnerability).
- **Line 120:** please provide (ranges in) average precipitation (surplus).
- **Figure 1b.** I would separate based on the four considered regions and not the provinces. Then, you can remove the map from Fig. 3.
- **Figure 1b.** you could show weather and precipitation stations on the map.
- **Line 131:** „overlapping timeseries were combined" and checked for consistency?
- **Line 134:** „precipitation" I would define precipitation ($P$) as you also do this for reference evapotranspiration ($ET_{ref}$). You could even go for $E_{ref}$, to be consistent with HESS guidelines.
- **Line 197:** I cannot judge what a negative value of f is and $-0.05 < f < -1.95$ is more concise.
- **Line 205:** would split these 44 further in groups with outliers (1) abnormal long-term behavior (2) or both.
- **Line 211:** „some errors" this is not very precise…
- **Line 224:** I do not like the use of the term mean error (ME) to describe bias. Why not simply refer to it as (mean) bias?
- **Line 225:** „the mean error" you just introduced an abbreviation for this term. Use it.
- **Line 226:** „mean(simulated - observed)" do not really like this. Why not use proper symbols (e.g. $G_{obs}$ or $GW_{obs}$) and put them into an equation.
- **Line 226:** why was rank correlation not considered. For the SGI, especially the non-parametric one, it does not really matter if you get the absolute values right but rather the ranking of values is important.
- **Line 253:** but non-parametric approaches can have greater uncertainties (Tijdeman et al., 2020).
- **Line 254:** „aggregated" see above.

- **Line 258-260.** I like this note and support the use of the SGI. But just wanted to mention that, if the probability is not important but just the rank, why do the probability transformation? Why not simply use ranks? Or, to take into account differences in record length, percentiles expressing the historical non-exceedance frequency in a given time period (Tijdeman et al., 2020)?
- **Table 1:** „dryest" driest.
- **Line 273:** „prediction". I do not think it is really prediction but more evaluation over the uncalibrated period.
- **Line 282:** „mean error ME" just ME

- **Line 291:** „the decay parameter of the noise model" I would prefer that this is briefly explained here so the reader does not necessarily have to go back to the mentioned study to understand the outcome of this equation.
- **Line 308:** „over-smoothing" to what does this refer?
- **Table 2: TP, TN** etc. are nowhere defined!
- **Line 320:** „20 % error" is this the average? Provide ranges?
- **Table 3** "20p" is not needed and neither is (mod-obs). Precision of units is not consistent, with some having only 1 digits behind the point and others 3.
- **Table 3.** Please add rank correlation.
- **Figure 3.** I don't like the map here (better in Fig 1), but if it is kept, I would rather show regions instead of center points.
- **Figure 4.** Nice! Just density instead of "kernel density" on the y-axis.
- **Figure 5 and 6.** Ok, but in my opinion, differences between regions / subsets of wells do not come out strongly from the maps (e.g difficult to recognize the different symbols). What could work is adding density functions split by region or split by well subset (deep, atypical etc.).
- **Line 359.** It should be emphasized that response time is not typical for the drought of 2018 but rather for the entire timeseries.
- **Figure 7.** Adding density plots (like Fig. 4), splitting response time by region or type of well (atypical etc.), might reveal some interesting results.
- **Figure 8.** Some redundant text in caption e.g. what the axis describe.
- **Figure 9.** comparison with simulations is difficult, not only because they are on another page but also because 2018-11 is only shown for the observations. Comparing density functions (possibly grouped by the different type of wells) of simulated and observed SGI might be a suitable alternative.
- **Line 392-394:** it is not clear where these numbers come from.
- **Table 4.** MAE etc. should be defined in the methods.
- **Table 4, line 401-405.** But does the performance maybe relate to well type (typical, etc)?
- **Line 459:** the comparison with Van Loon is not fair as the latter study compares the overall correlation between SGI and SPI / SPEI (not just during few drought months).
- **Line 490-491:** I really do not like this generalization (see above) and more analyses is required to make this claim, even within the discussion.
- **Line 500:** and the probabilistic uncertainty in the SGI estimation itself.
- **Line 509:** 30-year records in certain months. If you want to make a statement of breaking 30-year records overall, you should compare e.g. annual minima or duration.
- **Line 518-526:** the discussion in this paragraph, e.g., the importance of winter recharge, implies the importance of considering initial conditions. This contradicts some that is discussed before.

**References:**

Bakke, S. J., Ionita, M., and Tallaksen, L. M.: The 2018 northern European hydrological drought and its drivers in a historical perspective, Hydrol. Earth Syst. Sci., 24, 5621–5653, https://doi.org/10.5194/hess-24-5621-2020, 2020.

Bloomfield, J. P. and Marchant, B. P.: Analysis of groundwater drought building on the standardised precipitation index approach, Hydrol. Earth Syst. Sci., 17, 4769–4787, https://doi.org/10.5194/hess-17-4769-2013, 2013.

Brunner, M. I., Liechti, K., and Zappa, M.: Extremeness of recent drought events in Switzerland: dependence on variable and return period choice, Nat. Hazards Earth Syst. Sci., 19, 2311–2323, https://doi.org/10.5194/nhess-19-2311-2019, 2019.

Kumar, R., Musuuza, J. L., Van Loon, A. F., Teuling, A. J., Barthel, R., Ten Broek, J., Mai, J., Samaniego, L., and Attinger, S.: Multiscale evaluation of the Standardized Precipitation Index as a groundwater drought indicator, Hydrol. Earth Syst. Sci., 20, 1117–1131, https://doi.org/10.5194/hess-20-1117-2016, 2016.

Tijdeman, E., Stahl, K., and Tallaksen, L. M. (2020) "Drought Characteristics Derived Based on the Standardized Streamflow Index – a Large Sample Comparison for Parametric and Nonparametric Methods." Water Resources Research, https://doi.org/10.1029/2019WR026315

Van Loon, A. F., Kumar, R., and Mishra, V.: Testing the use of standardised indices and GRACE satellite data to estimate the European 2015 groundwater drought in near-real time, Hydrol. Earth Syst. Sci., 21, 1947–1971, https://doi.org/10.5194/hess-21-1947-2017, 2017.

---

## Author Comment (AC1)

**Response to reviewers' comments**

„Spatiotemporal development of the 2018-2019 groundwater drought in the Netherlands: a data-based approach" by Brakkee et al.

We want to thank all four reviewers for their interest in our study and their thorough and insightful feedback. As much of the feedback overlapped between the reviewers, we would like to give a general response first and outline the main revisions planned. Then we will discuss all major comments of the four reviewers individually.

From the reviewers' comments, we see two major points emerge that encompass much of the feedback:

1. **Better justification and explanation of methodology for data validation and time series modelling.**
   For the data validation methods section, the reviewers point to choices for parameters and thresholds that need better justification and explanation. We will therefore go through this section to ensure all choices are founded. Part of these parameters were derived from literature or knowledge of site conditions and for these we will add the appropriate references and explanations. A few of the parameters used in the data validation were chosen by tests on the dataset in an initial exploratory analysis. We plan to include a test of the performance of the validation method (as Table 2 in the manuscript) with different values of these parameters to evaluate their effect. In addition, we will add figures of the tested time series to show what the method means visually, probably as an appendix. We think this will provide the justification needed.
   For the time series modelling methods section, several of the reviewers ask to provide more information on the precise model setup, for which we now refer mostly to the paper by Collenteur et al. (2019). We will add a paragraph explaining how an impulse-response time series model (set up through Pastas) works, the different parameters and the settings used. An additional paragraph may be appropriate for this, but if the information needed to allow reproduction of the results is too lengthy, we will add it as an appendix. In addition, two of the reviewers have questions on the effect of linear versus nonlinear recharge modelling in our study. Comparing the two was not an aim of our study and we have not tested the nonlinear models sufficiently to draw any conclusions. We will therefore remove the statement on nonlinear modelling in Line 172-174 and discuss the effect of linear versus nonlinear modelling in the discussion section, as we agree this is indeed a very important topic for groundwater drought research.
   At the start of the methods section, we will add a definition of what we mean by 'usefulness' (research aim 1, line 105-106) and what components it consists of. This is now implicitly described in the introduction and methods sections; adding an explicit explanation will help to justify the choices made in the methodology.
2. **More focus in the topics covered.**
   From the reviewers' comments we get the feeling that the paper would benefit from skipping one or two of the themes discussed, and covering the remaining themes more thoroughly by adding some missing pieces (as explained above). The reviewers suggest the analysis and discussion on prediction performance is not essential to the study (Reviewer #1 and #3). We plan to skip drought prediction performance as a separate research topic, by moving the performed analysis on 'prediction performance' as an additional element to the model evaluation, as suggested by Reviewer #1. We will also remove the analysis of initial conditions (line 406-411) and remove most of the discussion on drought prediction performance from the discussion section, instead adding a short discussion on the potential of TSM for prediction and what clues our research gives for that.

Based on this 'red line' in the feedback received from the reviewers, the main revisions we plan to make are:

Main revision #1 Define what we mean by "usefulness" and what components it consists of.

MR #2 Add explanations and references to parameters, thresholds and other choices in the data validation section.

MR #3 Add a parameter test for the validation method, testing the effect of the parameters on error identification.

MR #4 Add figures in an appendix of series with different kinds of visual errors and the three different series types.

MR #5 Add a paragraph with explanation of impulse-response TSM and used model setup in Pastas; include this in an appendix if it becomes too lengthy.

MR #6 Remove prediction as a separate research topic.

MR #7 Add short discussion on the limitations of linear recharge modelling for drought conditions in the discussion section.

Below we reply to all non-minor reviewer comments and explain how we will address them through the above main revisions and other smaller revisions. The reviewers' comments are shown in *italics*, our responses are shown in plain text, and planned revisions in blue. The reviewers' minor comments will of course also be dealt with as we revise the manuscript.

**Reviewer #1**
**Methodological aspects**

*- Line 159: What is „short-term" (days, months, years?). And why are these not relevant. Please explain.*

Drought is generally considered to occur on timescales of months to years (van Loon, 2016). By 'short-term' disturbances, we mean disturbances on much shorter timescales of days to weeks. A week-long lowering of the water table for building works, as an example of a short-term disturbance, will generally not lead to drought impacts on nearby crops. Such variations also disappear mostly in the aggregation to monthly values we and many other groundwater drought studies use (e.g. Bakke et al., 2020; van Loon et al., 2015).

Revision: add explicit reference to timescales of drought and deviations in this section, referring to earlier studies.

*- Line 162: Why are long term disturbances relevant for drought studies? Please explain.*

If we define drought as 'a lower water level than normal' over a long period (months-years) (van Loon, 2016), then any sustained deviation in the water table is relevant, whatever driver causes it (van Loon et al., 2016). For example, sustained periods of increased water abstraction will cause sustained lower water levels than normal, and thus drought and drought impacts, even when the deviations are not expected from rainfall and evapotranspiration alone.

Revision: add an example of a 'long-term deviation' in this section.

*- Line 166: Why are time series with long-term changes less reliable for drought analyses (also given the prev. point). Or does this refer to their simulations?*

Here we refer to those series with a long-term trend or other long-term deviations from the pattern expected from rainfall and evapotranspiration. These deviations may be real, caused for example by

water management changes, but may also be caused by instrument drift or e.g. errors in the ground surface level metadata (Post & von Asmuth, 2013). Because of this possibility of an error causing the deviation, these series are less reliable.

Revision: add examples of real and erroneous long-term deviations in this section.

*- Line 170-71: I would prefer a bit more detailed description of the PASTAS approach here (I can read it from the mentioned publication, but a short summary of the approach and the used parameters would be easier for readers that do not want to go over all the details).*

We see that we have indeed been too brief in the model description. We will include an extra paragraph or appendix explaining the impulse-response approach and the used model setup.

Revision: add explanatory paragraph (main revision #5).

*- Line 173: describe the used non-linear approach in more detail (or delete / move to discussion that you tested a non-linear approach and how).*

The first functionalities for nonlinear recharge modelling have been included into Pastas towards the end of the period we performed the research. We have only done a very brief test of this functionality to see if it might have very large benefits for our case, but were not able to reach conclusive results on this. We therefore agree it is better to remove this comment from the methods and shortly discuss the nonlinear modelling in the discussion section.

Revision: remove nonlinear recharge modelling from methods and discuss its potential in the discussion (main revision #7).

*- Line 176: why a cutoff at >20 cm. Is the reason that GW head can exceed surface level related to inaccuracies in the used elevation data?*

The first cleaning step in line 176 is indeed meant to remove those data points that seem inconsistent with the elevation metadata of the well filter, which can be due to errors in the measurements or in the metadata. In parts of the study area, groundwater levels are shallow and also temporary shallow inundation is possible. Therefore, groundwater levels slightly above the surface are not necessarily wrong (see also Leunk, 2014). Deep inundation is less likely for most locations where groundwater level is measured as it would occur only by e.g. river flooding. We chose the 20 cm as a boundary between 'shallow' and 'deep' to remove the most unlikely data points without filtering out too much in advance.

Revision: add explanation for the 20 cm cutoff.

*- Line 177-178: the use of thresholds like 20% of data and 50% of the measurement ranges should be better justified or evaluated. Also, by focusing on both the wet and dry outliers at the same time, you might either: disregard some GW wells that are suitable for drought but have a lot of variance in the wetter domain or include some less suitable wells that have a high variance in the dry domain and a low variance in the wet domain. This should be discussed.*

The justification of the used thresholds in the data validation will be improved by main revisions #2 and #3. As for the removal of dry and wet outliers at the same time: this data validation step (step 2, Line 177-178) is meant to remove those measurements that are so far from the normal range that visual inspection would easily identify them as errors (the thresholds were determined by visual checking of series). In this step no wells are discarded yet, only individual measurements; it only serves to improve the model performance in further steps. We therefore believe no separate treatment of wet and dry outliers is needed. We will, however, include some figures of series in an appendix to illustrate this step.

Revision: examine the effect of the thresholds in the validation parameter test (main revision #3); add figures with examples of far and near outliers in the appendix (main revision #4).

- *Line 180: what is „basic model settings of PASTAS"*

Revision: explain the basic settings used in an extra paragraph or appendix (main revision #5).

- *Line 185: better explain A, f and d, and what these metrics mean.*

Revision: we will include these parameters in the explaining paragraph/appendix on the model approach (main revision #5).

- *Line 202 (and other places): The term „aggregated" alone can mean various things. Be precise.*

In all cases in the article, spatial and temporal aggregation has been done by taking averages, which we will indeed specify in the text.

Revision: Specify throughout how aggregation is done.

- *Line 205: why randomly select 120 GW timeseries and then select 56 from those 120. Why not just select 56 randomly?*

This was done to ensure both a good cover of the dataset (by random selection of 20 points from each province) and ensure a test set that included as many different types of errors as possible (by selecting from the first selection series with different kinds of visually clear errors and without those). We will explain this in the text.

Revision: explain approach for selecting test series in line 205.

- *Line 205-206: it would be helpful to show some exemplary timeseries with and without visual errors, preferably in the results (maybe in the supplement).*

We agree that some figures would help to understand the validation approach, as other reviewers have also suggested. We will add figures of visually erroneous and clean series in an appendix (Revision #5).

Revision: figures of visually erroneous and clean series in appendix (Revision #5).

- *Line 209-210: please provide abbreviations of true positive etc. here. Otherwise, few people will understand table 2.*

Revision: include abbreviations.

- *Line 218-219: please provide more detail about the PASTAS model as applied in this study here or in the previous section.*

See main revision #3.

- *Line 232: what are these „practical consequences"*

By the "practical consequences" of a model error, we mean that if the simulations would be used for water management decisions or impact studies, a certain error in simulated groundwater levels could lead to wrong decisions or conclusions depending on the local situation. If groundwater levels are normally already 5 meters below the surface and out of reach of most plants, an error in the simulated water level of half a metre will not lead to substantially wrong conclusions on the drought effects that are to be expected at that site. If levels are normally close to the surface, such a drop may have severe consequences for the vegetation. Therefore, small errors in model simulations will more easily distort the expected drought impacts in areas with shallow water levels.

Revision: refer to "drought impacts" rather than 'practical consequences' in line 232 to make this point more clear.

*- Line 238: which distribution is used for the SPEI?*

For the SPEI, we used a normal distribution for transformation. In the original paper (Vicente-Serrano et al., 2010) a log-logistic distribution is used, but we stayed with a simpler distribution as the SPEI data are not used quantitatively for analysis but only for graphical plotting.

Revision: include SPEI distribution in line 238.

*- Line 240: why not simply the mean of all stations in a region?*

Reference evapotranspiration was available only from 15 stations, and the set of stations within a zone did not always cover the zone well (e.g. no stations in one half of the zone, or stations at the very edge near the sea). The average of the three stations closest to the midpoint therefore gave a more representative image of ETref in the zone.

Revision: note the number of ETref stations again in this section to make this more clear.

*- Line 250: „probably insufficient to do this reliably" either test this or delete.*

Other studies have elaborately tested the number of data points needed to fit reliable probability distributions for groundwater level and other hydrological series. Link et al. (2020) find that in most cases more than 100 data points are needed for fitting reliable parametric distributions on hydrological series. Therefore, we think the statement made here that 30 years of monthly data is likely insufficient for reliable parametric distribution fitting is reasonable, even though not tested by ourselves.

Revision: add reference to Link et al. again in Line 250.

*- Line 256: the original SGI paper of Bloomfield and Marchant (2013) used 1 / (2n) and not 1 / n. Please explain why you differentiate.*

We thank all reviewers for their precise eye to spot this error for us. This is a typo, and in the scripts the 1/(2n) has been used as by Bloomfield and Marchant.

Revision: correct the 1/(2n) error.

*- Table 1: values are not correct. E.g., if the lowest SGI is based on 1 / n (1 / 30), the lowest SGI value would be >-2.*

See previous comment.

*- Line 269-272. When comparing observed vs simulated SGI, did you match their record lengths. For example, if the simulated SGI covers 1990-2019, but the observed SGI 1993-2019, did you recalculate the simulated SGI for 1993-2019?*

This is a good point that deserves some further investigation. We did not recalculate the SGI for the simulated series, so that indeed the periods might differ slightly between the simulated and measured series. This could hamper the comparison over summer 2018 in two ways. Firstly, severe summer droughts could have occurred in 1990-1992 that are included in the simulated series but not in the measured ones. However, no substantial drought periods occurred in this period except for the winter and early spring of 1991. Being in a different season, this will not affect the drought index calculated for the summer period.

The small differences in covered periods (because of missing data in 1990-1992 or missing values in the rest of the period) could also affect the comparison through a difference in the number of samples determining the SGI values. The SGI values are defined by the standard normal distribution values

corresponding to p=1/(2n) to p=1-(1/(2n)). Therefore with 30 monthly values (n=30) the driest three years have SGI -2.13, -1.64 and -1.38; while with 25 monthly values (n=25) these are -2.05,-1.55 and -1.28. Looking at Figure 8 in the manuscript, the deviations visible in the lowest end of the drought spectrum cannot be explained by this difference in *n* alone. Making the same plot as Figure 8 but calculating SGI for measured series only if n=30 (all periods the same) gives the following graph:

[Figure]

The limited number of points that remain hint to the same patterns as Figure 8 in the manuscript, with simulated series showing largely the same drought behaviour as measured series except for a certain overestimation in the driest ranges.

All in all, we are confident that the small differences in periods in the measurement-simulation comparison for some of the series does not affect the results substantially. We will include the above explanation shortly in the text to justify this.

Revision: add to line 269-272 that record lengths were not matched and shortly why we think this does not affect results.

*- Line 294: did you combine simulated and observed data here?*

Yes, we did. We artificially created a 'predicted' series (simulating the drought period forward from spring 2018) and a 'true' series (with the prediction period replaced by the measurement) as the 'ideal reference' to compare the predicted series to. As explained above, we will remove the prediction as a separate topic and include the analysis in section 3.4 under the topic of simulation reliability.

*- Line 360-361. This should be presented in the method section and more clearly explained.*

We agree with this suggestion. We will explain the calculation of the response times in the method section, including the formula used.

Revision: include method for response times in methods section.

In addition, we will adjust the "minor comments" mentioned by Reviewer #1.

**Reviewer #2**
**Specific comments**

*Title/focus of the manuscript – The text itself is focussed on the TRM-based approach, while the title is drawing more attention to the drought analysis. I would reconsider the title to reflect better on the*

*manuscript's content. I am also not sure about the use of the term "data-based" approach (most studies are based on data?).*

Indeed, the focus of the study is on evaluating the use of TSM methods for drought analysis, with the Dutch 2018-2019 drought as a test case. We agree that the title should be adapted to better reflect this focus. The term 'data-based' was chosen initially to distinguish this study, which is based fully on groundwater data and directly derived TSM simulations, from studies into groundwater drought with physically-based models.

Revision: change title to "Time series modelling for improved understanding of spatiotemporal groundwater drought development"

***Figures*** *– The figures are easy to read and support the conclusions. On Figure 9 though, it would be more useful to present the SGIs of the measured time series for the same months as for the simulated time series (Figure 5).*

This is a very good point, and we will indeed adapt Figure 9 to show the same months as Figure 5 to allow easy comparison.

Revision: show April, July and October in Figure 9.

***Methods*** *– The methods are clearly explained in principle, but in some instances, it would be helpful to elaborate a bit on why specific cut-offs were chosen. Please add some more information e.g. on:*

*L165 – 'one or a few', Did you use a specific threshold for 'a few'? If yes, this should be added.*

Here we are trying to define generally what we mean by an 'outlier'. The actual thresholds used for identifying these outliers are mentioned in line 177-183. We agree that it would be better to define 'outliers' more formally here as measurements far from the expected behaviour over short periods of time (in our case days-weeks, see comments for reviewer #1) (see also Peterson et al., 2018).

Revision: provide definition of "outliers" with literature reference.

*L175 – Please elaborate on which basis you decided on the cut-off at 20 cm.*

In the study area groundwater levels slightly above the surface (shallow inundation) are generally possible, but measurements showing a groundwater table far above the surface are unlikely and probably point to errors. The boundary of 20 cm between 'shallow' and 'deep' inundation was chosen as an estimate to not filter out too much in advance but still remove the most unlikely data points.

Revision: add short explanation on the 20 cm cutoff.

*L183 – As in L175, a bit more explanation on how procedures (in this case repeating outlier removal twice) would be helpful.*

As explained under main revisions #2 and #3, we will indeed add explanations and references to the parameters and thresholds used in the data validation where possible, and evaluate the others in a parameter test on the data validation. In this case, we found it was necessary to repeat the outlier removal because far outliers sometimes distorted the model fitted, so that smaller outliers could not be detected. A second step then allowed to identify these smaller outliers. In principle, one could continue these iterations until no outliers are present any more (e.g. Peterson et al., 2018), but we found that with our setup, repeating too often could lead to removal of points that were visually not clearly erroneous, and therefore we kept the number of iterations at a conservative two.

Revision: explain why two rounds of simulation were used in Line 183; and more generally main revisions #2 and #3.

*L131-132 – Why were those series with > 10 years of data selected? I would be helpful to elaborate a bit the cut-offs. Do the 10 years refer to the amount of datapoints, or to 10 years of consecutive data? Did you also consider a maximum length of allowed data gaps?*

We chose the 10 years as a minimum to be able to fit time series models reliably. To fit an impulse-response model, a data series should encompass preferably several times the response time of the groundwater level to recharge input. Zaadnoordijk et al. (2019) use a minimum of 8 years as an experience-base rule of thumb for the Dutch situation for fitting impulse-response models; we chose a slightly longer period to also fit slower locations and encompass enough weather variability.

By 10 years we refer to 10 years of consecutive data with at least a monthly timestep, but with data gaps allowed. Series with long data gaps (>4 years) are filtered out in a later step (line 191).

Revision: give short explanation in the text.

*L204-206 – The selection of the time series could be explained a bit more in detail; e.g. why around 120 series were selected at first.*

Here we first randomly selected 20 points from each province (giving 120 series) to ensure we had a good cover of the dataset, and then selected from the first selection series with different kinds of visually clear errors and without those to make sure the test set covered the different kinds of errors in the data.

Revision: add an explanatory sentence on data selection to line 204-206.

*L191 – 4 consecutive years? Also add why data from June-August 2018 was considered as particularly crucial.*

The drought of 2018-2019 was at its most extreme in summer 2018, as explained in the introduction. As the aim of the case study was to map the development and distribution of this specific drought we chose to include only those series that had measured information of this most severe period, rather than interpolating this period with the time series modelling.

Revision: clarify this shortly in Line 191-192.

*L235 – Elaborate on why you chose the three-month-aggregated SPEI*

We chose the three-month aggregation because this most clearly showed the meteorological droughts at a time scale comparable to the variations in groundwater level. With shorter time scales, the SPEI series shows very short-term droughts and wet periods that do not give insight into the weather as a driving factor for groundwater drought, which was the aim of the SPEI timelines in Figure 3.

Revision: clarify shortly the match of SPEI with groundwater time scales.

*L255 – Why did you chose 1/n instead of 1/2n?*

This is a typo, and in the data analysis scripts the 1/(2n) has been used as by Bloomfield and Marchant.

Revision: correct the formula in Line 255-256.

*Table1 – Please add the reference for drought classification. It might be good to discuss in the text, why this classification was chosen. In other literature (e.g. Svoboda, M., Hayes, M., and Wood, D.: Standardized precipitation index user guide, World Meteorological Organization, Geneva, Switzerland, 24 pp., 2012.) drought periods are only defined if the index is continuously negative and reaches an intensity of -1.0 or less.*

We used the drought classification as introduced by McKee et al. (1993). We chose it because it has been frequently applied in other (groundwater) drought studies (e.g. Marchant & Bloomfield, 2018)

and may therefore be intuitive to many readers. Indeed, other classifications have been used as well. However, we use the classes themselves mainly to be able to give interpretable colours to display the variability in drought severity in the figures, and not to derive any drought statistics, so the classification is not of large importance for the results.

As for the consideration of a minimum duration with negative SGI, we did not consider this necessary to identify drought in the groundwater: as groundwater levels generally vary slowly, an SGI below -1 or less already signifies drought conditions that have accumulated over some time, and will generally not recover over a very short period (<1 month). Therefore, we consider the direct SGI a suitable and straightforward indicator of drought severity for our case.

Revision: include reference to McKee et al. in Line 260.

*L270 – Why January 1993?*

The aim of this analysis is to compare the SGIs derived from simulated series to SGIs derived directly from measured series. This requires the measured series to ideally cover the same 30-year period as the simulated series. However, taking the period 1990-2019 as a hard criterion leaves only a very limited number of series for comparison. We therefore allowed series to be slightly shorter. As explained in more detail for reviewer #1, allowing the series to miss at most 3 years of data in the beginning and at most 5 missing data points for the individual months has little influence on the resulting comparison with the simulated SGIs, while still allowing a good number of series for comparison.

Revision: explain shortly why this limit was chosen in the text.

*L272 – Please add the type of regression analysis*

Revision: note in text that Spearman's regression was used because of the ordinal data type.

*L321 – Please reiterate based on what the dry, normal and wet conditions are defined.*

The definitions of 'dry', 'normal' and 'wet' conditions are shown in Table 3 (0-20$^{th}$ percentile, etc.), but we will indeed include them in the text as well.

Revision: add the percentiles used for dry, normal and wet in the text.

We will also correct all "minor specific comments" and "technical corrections" brought up by Reviewer #2.

**Reviewer #3**
*Applying impulse-response time series modelling on so many groundwater observation wells in order to analyze the drought pattern on a regional scale is an impressive work. However, I am not really convinced by the necessity and the advantages of the presented approach in the analysis of a drought event. This is partly connected to missing or sometimes rather vague explanations/justifications partly to the approach itself. To overcome the first issue, it might be advisable to focus either on a) the analysis of the drought event or b) the drought prediction.*

We will ensure the consistent justification and explanation of methodological choices through Revisions #1 to #3, by adding literature references and explanations where possible, and adding an evaluation test of the most important parameters/thresholds used in the validation. We plan to remove the prediction topic as a separate theme, and fit it into the model reliability/evaluation theme (Revision #7), improving the focus of the study.

***Main Comments***

*In general, I think impulse-response time series modelling can be a useful tool in the analysis of groundwater systems. As you point out, the approach is able to give site-specific information (e.g. response time) and might also be a powerful tool in forecasting groundwater heads. However, reading the manuscript I am still not convinced by the advantages of this approach in connection with the SGI in the analysis of a drought event. Most of the following statements are related to the question of how much information you gain or lose before calculation SGI values.*

We believe the value of TSM data preparation for drought specifically is that for drought analysis long series are needed, which are often available for only a limited number of locations; therefore for drought research specifically TSM can be helpful to increase the spatial cover.

We fully agree that TSM-based data preparation (cleaning, simulation) for groundwater drought involves both gains and losses. We evaluate both of these in our study. The loss of information caused by TSM simulation consists of errors and real external impacts that are (partly) filtered out in the simulation. This loss of information is quantified in Table 3 and Figures 8 and 9, where drought assessment on simulated and non-simulated series are compared. The increased scatter in the measurement-based maps indeed points to some loss of information especially during severe drought. On the gains side, Figure 9 shows the gain reached by TSM data preparation, as the image of the drought situation Figure 9 provides based on non-simulated data is much less complete than that in Figures 5 and 6 in which many more points can be used. The advantages and the weak points of TSM data preparation for drought studies are discussed in section 5.1.2. Based on these, we argue that TSM data preparation, especially if further developed, can be very useful in many groundwater drought studies, but this choice should be made for every situation individually based on the goal and the used data.

*L177: Why do you use 20% as a cutoff, please elaborate.*

See main revisions #2 and #3. This cutoff was determined by trial and error on the test series, but we will evaluate it more fully in the validation parameter test (main revision #3).

Revision: include 20% in data validation test.

*L179: Data cleaning step 2 in combination of using the model results of PASTAS based on time series of precipitation and evapotranspiration to remove outliers homogenizes the time series. Don't you lose specific features in the time series especially those related to dry conditions? Is this what you mean by 'over-filtering' (L211)?*

Indeed, the data cleaning and simulation to some extent homogenise the series. In the data preparation for drought analysis, a balance needs to be found between removing errors and short-term disturbances and increasing the number of series useable for analysis on the one hand; and inevitably losing some potentially real information on the other hand. This balance is discussed in line 146-167 and 427-441 and section 5.1.2. We evaluate the loss of information caused by data cleaning and simulation in Table 3 and 4 and Figures 8 and 9. These show that indeed some information appears to be lost through cleaning and simulation; however, the overall conclusions resulting from the analysis on the development of drought still appear robust (line 407-417). As visible in Figure 9, not simulating the series but relying on the direct measurement series would have resulted in much fewer series for the drought analysis, and therefore also potentially incomplete conclusions on the drought development. We therefore believe we have paid due attention to the loss and gain of information by time series simulation and cleaning in the analysis and discussion. Further developments in time series model cleaning methods may further improve the balance between loss and gain, as explained in line 434-441.

*L196-200: What are the reasons for 'atypical behaviour'? Could site-specific characteristics, i.e. hydrogeology, play a role? In general, I think it would be good for the reader to see either some of the original groundwater head time series or the model outcome.*

As explained in line 196-200, series were defined as 'atypical' if they could be reasonably modelled by rainfall and evapotranspiration, but showed either a trend or one of the parameter values close to the bounds of its physically realistic range (physically realistic parameter bounds are included in the Pastas package, see Collenteur et al. 2019, p. 880). Trends in the series may be caused by changes in water management or by water abstraction; but may also be caused by e.g. instrument drift (Post & von Asmuth, 2013). Atypical parameters signify that the model imposed on the data – a wave-type delayed response of groundwater to rainfall and evapotranspiration - may not be suitable for the given series, potentially because some other process is influencing the series. Hydrogeological site conditions may play a role, but looking at the maps of e.g. Figure 5 and 7, the 'atypical' locations are located rather randomly in space rather than concentrated in some regions as would be expected for large-scale hydrogeological impacts.

We agree that showing some of the actual series is a valuable addition. We will include some figures of the different classes of series in an appendix.

Revision: include figures of normal, atypical and deep locations in appendix (main revision #4).

*L201: Why do you use 60%? It seems to be very low. The distribution (histogram) of the models' EVP would be interesting. It might also be interesting to compare time series with different EVPs, e.g. 60% and 80%. Where are the differences – extremes/timing etc.?*

Various criteria on the EVP are used in time series literature, also depending on the goal of the study. Zaadnoordijk et al. (2019) use 30%, while Leunk (2014) uses 70%. We chose the criterion slightly low because we wanted to take care not to remove too many series. We will include the EVP criterion in the validation parameter test (main revision #3). What the 60% criterion means visually will also be shown by the figures of normal, deep, unusual and discarded series we will include (Revision #5).

Revision: main revision #3 and #5.

*L254: How do you aggregate the daily data? Do you really need a time series lengths of 30 years in a daily resolution to calculate the SGI values for 2018? Is it not efficient enough to use time series with a coarser temporal resolution and a time series length shorter than 30 years? How many measured data sets out of the 2722 fulfill weaker requirements in terms of measurement frequency and length, e.g. at least weekly data from the past 20 years? How would the SGI values (using normal score transformation) and their spatial distribution look like for these time series. How much extra information do you get of using your approach? This kind of analysis would be interesting in order to show the advantages of this approach.*

The daily data were aggregated by taking monthly means. The minimum of a 30 year record to identify droughts from a hydrological series was originally proposed by McKee (1993) and has since then been often used as a standard in drought research (van Loon et al., 2016). As 'drought' is defined as a deviation from some long-term 'normal condition', enough data must be available to estimate this normal condition; 30 years is an often used period for this groundwater research (e.g. Ritzema et al., 2018) as well as in climate research. In addition, for calculating drought indices some parametric or non-parametric distribution needs to be fitted on the data, for which 30 years may actually already be short (Link et al. 2020). Therefore, we consider 30-year data series an important starting point for the drought analysis. We do not take daily data as a prerequisite to use a series; as explained in line 128-129, the data are partly bi-monthly and partly daily-step.

The extra information gained by lengthening series rather than using the raw dataset is visualised in Figure 9. Figure 9 shows the SGI map that would result from using non-simulated data only, with the prerequisite of at least a 27-year long series and at least 25 values for an individual month to show the SGI. This still gives over 500 data points, but shows much less spatial detail than is possible by lengthening series. We therefore think that for the goal of obtaining a regional image of drought

development series lengthening does provide important extra information, as also discussed in the discussion section.

Revision: include how aggregation was done; include references for 30 year minimum in text.

We will also adjust all "secondary comments" by Reviewer #3.

**Reviewer #4**

We are pleased to have received this additional comment from mr. Collenteur, as the Pastas package has been of great use in our study. We have noticed that functionalities for nonlinear modelling and to calculate SGIs have been added to Pastas since we performed this study, which will be very interesting for further drought research.

**Effect of use of linear model**

*An important assumption underlying this study is that a linear recharge model can be used to accurately simulate the effects of precipitation and evaporation on the GWL. Previous studies have shown (e.g., Berendrecht et al., 2005; Peterson and Western, 2014; Collenteur et al., 2020) that this assumption may not always be valid, particularly during drought periods when non-linear unsaturated zone processes become more important (e.g., evaporation limited by the availability of soil moisture). This is particularly important because these droughts events are the periods of interest in this study. This model deficiency may partly explain the large RMSE and ME values in Table 3 of the manuscript and the results shown in Figure 8. The linear model could still be an appropriate choice here, but a justification of this assumption is required in my opinion. The impact of this assumption on the estimated SGI values and the results in general could also be discussed in the discussion section (e.g., in lines 450-455).*

We have indeed considered the limitations of linear recharge modelling, as indeed during extreme drought periods nonlinear recharge processes may become especially important. The main reason we used only linear modelling is that it has often been successfully used for Dutch locations (e.g. Zaadnoordijk et al., 2019 and work by von Asmuth); and because nonlinear recharge modelling was not available yet in Pastas when we performed the research in spring 2020. As we saw the first nonlinear functionalities had been introduced towards the end of our study, we briefly checked if it might have very large benefits for our case, but were not able to reach conclusive results on this; as the comparison was also not a specific aim of the study, we stayed with the initial linear models. However, we fully agree that nonlinear recharge modelling is an important topic to explore in further research, especially in relation to drought. We plan to remove the comment on nonlinear modelling from line 172-174 and discuss the shortcomings of linear recharge modelling in the discussion. The differences in model performance in dry periods (table 3, Figure 8) indeed suggest that some drought-related processes may be missed.

Revision: discuss (non)linear recharge modelling in the discussion (main revision #7).

**Uncertainty in time series modelling and its impact on the SGI**

*Time series models are used to obtain regular GWL time series, comparable to the approach presented by Marchant and Bloomfield (2018). In that study the uncertainty of the simulated GWL was also quantified and used to compute the uncertainty of the SGI values. I think it would be interesting to do this in this study as well (or discuss why this is not done), given that, despite generally good fits, the simulated GWL time series and thus the SGI may still have considerable uncertainties.*

Marchant and Bloomfield (2018) have used a different method than we have used to fill gaps in series, as they use TSM to interpolate and extrapolate missing periods rather than simulating whole series. The uncertainty quantification they perform is therefore not directly applicable to our method. However, we have looked into the reliability of the GWL simulations and the resulting SGI values

through the model errors with the drought period inside or outside the calibration period (table 3 and 4); and how this propagates to resulting SGI values (table 4, Figure 8, Figure 9).

**Reproducibility of results**

*Some of the claims made in the manuscript directly depend on how the time series modeling was done, which is very briefly described in section 3.1 and 3.2. From the information contained in the manuscript it is not possible to reproduce the results from this study, to verify any of these claims. I therefore think some work is required to improve the reproducibility of the presented work. This could be a much more detailed description of the modeling process (e.g., settings, model structure, calibration settings, software versions), but perhaps an easier way to do this would be to upload all scripts and data (if allowed) to an online repository (e.g., Zenodo) and assign an DOI. This would enable other researchers to build upon this work more easily and make it a more valuable contribution.*

As for the data used, all the groundwater data are publicly available from the national groundwater database DINO ([www.dinoloket.nl](www.dinoloket.nl)) and the weather data from KNMI via [http://projects.knmi.nl/klimatologie/daggegevens/selectie.cgi](http://projects.knmi.nl/klimatologie/daggegevens/selectie.cgi) (see Line 570-574).

As for the data processing and modelling, we agree that the methods should be made reproducible as much as possible. We are investigating the options of sharing scripts and workflows directly in an online database, but this may not be possible yet for the scripts of this specific study (due to current administrative limitations). We will add a more detailed description of the settings of the time series modelling in the methods section or in an appendix (main revision #5). All scripts used for the study can be obtained from the authors by anyone interested, as indicated in Line 570.

**Specific Line Comments:**

*L168. I would kindly ask the Authors to change "PASTAS" to "Pastas" throughout the manuscript.*

Revision: we will change PASTAS to Pastas throughout.

*L168-170. I think it would be good to rephrase this sentence, because it reads as if this was the goal of developing Pastas, which is incorrect. The use of (Python) scripting is what allows the models to be applied in larger workflows.*

Revision: rephrase sentence to "This is a package for analysis of hydrological timeseries developed by Collenteur et al. (2019) and allows impulse-response time series models for groundwater heads to be easily applied and combined into larger workflows."

*L170. It is unclear from the manuscript or the reference what "the basic settings of Pastas" are. It should either be described in the manuscript, or the scripts and data can be provided in an Appendix or external repository. Sharing the scripts would help in improving the reproducibility, without requiring the Authors to go into details about the modeling in the manuscript itself and increasing its length.*

We will add the necessary details of the model setup inside the manuscript or an appendix (main revision #5).

*L172. Pastas has multiple non-linear recharge models, based on Berendrecht et al. (2005) and Collenteur et al. (2020). It is not clear which method has been tried here, but this would be valuable information. Moreover, the statement is not supported by any data presented in this manuscript, so it is hard to verify such a general statement. Since the non-linear models have more parameters to fit the model to the data, compared to the linear model, I find the finding somewhat surprising. However, it could be the case that the linear model really does work better, perhaps because evaporation of groundwater occurs at most monitoring wells (which would be an interesting finding in itself).*

As explained above, we have only done a very limited test of the nonlinear recharge models in Pastas and this is a topic to be further explored in future research. We will remove this comment on

nonlinear model testing from the methods and discuss the need for exploring nonlinear recharge models in the discussion section (main revision #7).

*L173-174. "Small minority" could be quantified (e.g., XX number of wells). Also, the current phrasing in this sentence seems to suggest that a non-linear model was used for some locations, but in that case no parameter "f" would be available for some models (L.185).*

See previous comment, we will remove this sentence. The presented results are all based on linear recharge models.

*L462. The fact that the drought was overestimated may also result from the use of a linear model, where evaporation is not limited by soil water availability and may ultimately lead to a decrease in simulated GWL.*

This is indeed a possibility, that further stresses the need to explore nonlinear recharge modelling further for drought studies. However, the model performance over the full period (Table 3) shows that groundwater levels in dry periods in general tend to be simulated (slightly) too high. This puzzled us and suggests that model setup may not be the only factor explaining the deviations in simulated drought in 2018-2019; or that extreme drought periods such as in 2018-2019 have their own unique dynamics. We will include the possible influence of linear recharge models in this section (main revision #7).

**References**

Bakke, S. J., Ionita, M., and Tallaksen, L. M.: The 2018 northern European hydrological drought and its drivers in a historical perspective, Hydrol. Earth Syst. Sci., 24, 5621-5653, 10.5194/hess-24-5621-2020, 2020.

Collenteur, R. A., Bakker, M., Caljé, R., Klop, S. A., and Schaars, F.: Pastas: Open Source Software for the Analysis of Groundwater Time Series, Groundwater, 57, 877-885, https://doi.org/10.1111/gwat.12925, 2019.

Leunk, I.: Kwaliteitsborging grondwaterstands- en stijghoogtegegevens. Validatiepilot; analyse van bestaande data. , KWR Watercycle Research Institute, Nieuwegein, The Netherlands, 2014.059, https://edepot.wur.nl/317083, 2014.

Link, R., Wild, T. B., Snyder, A. C., Hejazi, M. I., and Vernon, C. R.: 100 years of data is not enough to establish reliable drought thresholds, Journal of Hydrology X, 7, 100052, https://doi.org/10.1016/j.hydroa.2020.100052, 2020.

Marchant, B. P., and Bloomfield, J. P.: Spatio-temporal modelling of the status of groundwater droughts, Journal of Hydrology, 564, 397-413, https://doi.org/10.1016/j.jhydrol.2018.07.009, 2018.

McKee, T. B., Doesken, N. J., and Kleist, J.: The relationship of drought frequency and duration to time scales, Proceedings of the 8th Conference on Applied Climatology, 1993, 179-183,

Van Loon, A. F.: Hydrological drought explained, WIREs Water, 2, 359-392, https://doi.org/10.1002/wat2.1085, 2015.

Ritzema, H. P., Heuvelink, G. B. M., Heinen, M., Bogaart, P. W., van der Bolt, F. J. E., Hack-ten Broeke, M. J. D., ... and van den Bosch, H.: Review of the methodologies used to derive groundwater characteristics for a specific area in The Netherlands, Geoderma regional, 14, e00182, 2018.

Peterson, T. J., Western, A. W., and Cheng, X.: The good, the bad and the outliers: automated detection of errors and outliers from groundwater hydrographs, Hydrogeology Journal, 26, 371-380, 10.1007/s10040-017-1660-7, 2018.

Post, V. E. A., and von Asmuth, J. R.: Review: Hydraulic head measurements—new technologies, classic pitfalls, Hydrogeology Journal, 21, 737-750, 10.1007/s10040-013-0969-0, 2013.

Van Loon, A. F., Kumar, R., and Mishra, V.: Testing the use of standardised indices and GRACE satellite data to estimate the European 2015 groundwater drought in near-real time, Hydrol. Earth Syst. Sci., 21, 1947-1971, 10.5194/hess-21-1947-2017, 2017.

Van Loon, A. F., Stahl, K., Di Baldassarre, G., Clark, J., Rangecroft, S., Wanders, N., Gleeson, T., Van Dijk, A. I. J. M., Tallaksen, L. M., Hannaford, J., Uijlenhoet, R., Teuling, A. J., Hannah, D. M., Sheffield, J., Svoboda, M., Verbeiren, B., Wagener, T., and Van Lanen, H. A. J.: Drought in a human-modified world: reframing drought definitions, understanding, and analysis approaches, Hydrol. Earth Syst. Sci., 20, 3631-3650, 10.5194/hess-20-3631-2016, 2016.

Vicente-Serrano, S. M., Beguería, S., and López-Moreno, J. I.: A Multiscalar Drought Index Sensitive to Global Warming: The Standardized Precipitation Evapotranspiration Index, Journal of Climate, 23, 1696-1718, 10.1175/2009JCLI2909.1, 2010.

Zaadnoordijk, W. J., Bus, S. A. R., Lourens, A., and Berendrecht, W. L.: Automated Time Series Modeling for Piezometers in the National Database of the Netherlands, Groundwater, 57, 834-    843, https://doi.org/10.1111/gwat.12819, 2019.

---

## Referee Report (RR1)

Review of "Improved understanding of regional groundwater drought development through time series modelling: the 2018-2019 drought in the Netherlands" by Esther Brakkee et al.

The manuscript greatly improved. The methods are better explained, and methodological choices are better justified and evaluated with an elaborate sensitivity analysis according to multiple criteria. Also, results and conclusions are more complete, although I had two (optional) suggestions that might be worth showing or exploring a bit further. Finally, I had a couple of minor remarks that need to be addressed.

**Suggestions:**

- For the simulation performance (Table 5) you only use absolute evaluation metrics (RMSE and ME). However, your purpose (in this study) is to derive the SGI. For this SGI, it does not really matter if you get the absolute GW vallues right, especially if you use a non-parametric one. What matters is that you get the ranking right. Therefore, I would still like to see a relative evaluation metric, preferably spearman`s rank correlation. This could be added as extra collumm to table 5.
- You did a great effort of data validation and classification, and grouped groundwater series into „typical", „atypical", „deep" (4.1.1). Howerver, you then more or less pull all classes together again for the remaining of the manuscript, while information about the different subgroups might contain some important information. It might be worth to derive response time (Fig. 7) for each of the 3 classes, like you now also do with the density functions for the different regions (which is nice!). Also, model performace (Table 5) and SGI performance (Fig. 8) could be evaluated for each class seperately.

**Minor remarks:**

- Line 22: abreviation TSM not yet defined.
- Line 117: abreviations P and $ET_{REF}$ are not yet defined.
- Line 117: not mm **j**aar but mm **y**ear$^{-1}$
- Line 127: what does „consecutive" mean here. At least one obsevation in each month for a consecutive period of 10 years? Or?
- Line 128: brackets around (2018) can be removed.
- Line 128-130: Fig. 3 is introduced before Fig. 2.
- Figure 2. Nice workflow Figure! Possibly add an arrow pointing from step 1 to step 2 and one backward from step 2 to step 1, to indicate that you repeat these steps to improve models.
- Section 3.1: I wrote down a note after seeing your EGU presentation that I realy liked your expanatory Figure of the method. For me, such a Figure would be very helpfull to quickly understand the method and where which parameter is involved. But I realize that it would also take quite some more space, so feel free to ignore.
- Eq 1: ETref also a function of (t)
- Line 156: not Appendix A but Appendix A1
- Line 158: „five parameters" contradicts with the „four parameters" stated in Appendix A1.
- Line 159: „1 mm" is this 1 mm per time step? If so, please add.
- Line 173: appendix A3 introduced before appendix A2.
- Line 242: GWL not defined.
- Line 248: WTD already defined a few paragraphs above.
- Line 268: Why „for simplicity"?
- Table 1: Isnt $f$ a positive factor, give the minus sign in eq. 1?
- Line 304: correlation instead of „regression"?

- Line 400-404: I like the addition to Fig. 7, but think this deserves a little bit more attention in text.
- Appendix: Might consider moving some of those (e.g. A2 & A3) to the suplementary material to limit the length of article.
- Line 588-590: To save a few lines, you might just refence eq. 1 in the main text.
- Line 596-599: repitition.
- Table A1: Isnt $f$ a positive factor, give the minus sign in eq. 1
- Appendix A3. Nice examples!